# Fexinidazole results in specific effects on DNA synthesis and DNA damage in the African trypanosome

Kenna E. Berg[1], Indea Rogers[1], Hayley M. Ramirez[1], Julian Cornejo[1], Ignacio M. Durante[1], Galadriel Hovel-Miner[1,2]*

**1** Department of Microbiology, The George Washington University, Immunology, & Tropical Medicine, Washington DC, Washington, United States of America, **2** Department of Microbiology & Immunology, Stony Brook University, Stony Brook, New York, United States of America

* galadriel.hovel-miner@stonybrook.edu

## Abstract

Nitroaromatic drugs are of critical importance for the treatment of trypanosome infections in Africa and the Americas. Fexinidazole recently joined benznidazole and nifurtimox in this family when it was approved as the first oral monotherapy against Human African trypanosomiasis (HAT). Nitroaromatic prodrugs are bioactivated by the trypanosome-specific type I nitroreductase (NTR) that renders the compounds trypanocidal. The trypanocidal activity of nitroaromatic drugs has been proposed to arise from the formation of reactive oxygen species (ROS) and DNA damage. However, the specific cytotoxic effects of nitroaromatic drugs had not been thoroughly interrogated. Here we evaluate and compare the effects of clinically relevant anti-trypanosome nitroaromatic drugs using cell biology phenotypes including cell cycle progression, DNA synthesis, and DNA damage formation in the African trypanosome. We observed that fexinidazole induced cytotoxicity is distinct from related nitroaromatic drugs in its inhibition of DNA synthesis and the timing and magnitude of DNA damage formation. These findings highlight the relationship between nitroaromatic drug treatments, DNA damage formation, and ROS activation. Deconvolving the relationship between anti-parasitic drugs and the molecular basis of their cytotoxic outcomes will support future mechanistic understanding and enable improved drug design.

## Author summary

Human African Trypanosomiasis (HAT) is a devastating parasite infection that is largely fatal if left untreated. Treatment options against HAT are limited to only a handful of drugs, most of which are heavily burdened by complex treatment regimes, host toxicity, and emerging drug resistance. Fexinidazole provides new

**Data availability statement:** Raw data are included in the supplemental materials of the manuscript.

**Funding:** This work was supported by the National Institutes of Health from grant numbers R01AI170769 and R21AI174051, which were both awarded to GH-M. GH-M receives partial salary support from the above mentioned NIAID funding sources. The funders had no role in study design, data collection and analysis, decision to publish, or preparation of the manuscript.

**Competing interests:** The authors have declared that no competing interests exist.

hope in the treatment of HAT as a clinically approved oral monotherapy. However, little is known about how fexinidazole kills trypanosome parasites. Here we present the first comprehensive analysis of the trypanocidal outcomes of fexinidazole treatment. The data herein demonstrate that fexinidazole treatment causes an accumulation of DNA damage and a significant inhibition of DNA synthesis. While the precise molecular mechanisms underlying these outcomes remain to be elucidated, these findings provide critical new insights into fexinidazole's trypanocidal activity.

## Introduction

Fexinidazole has recently joined nifurtimox and benznidazole in a group of nitroaromatic compounds critical to the treatment of human trypanosome infections [1]. Trypanosomes are unicellular protists in the group kinetoplastida that cause the significant human diseases: African trypanosomiasis (*Trypanosoma brucei spp.*), American trypanosomiasis (Chagas disease, *Trypanosoma cruzi*), and Leishmaniases (*Leishmania spp.*). While these related human parasites differ in global distribution and disease progression, they are all transmitted by insect vectors and have an initial acute phase that can be followed by potentially fatal chronic infections. Human African Trypanosomiasis (HAT) is unique in that parasite replication is entirely extracellular and HAT infections are usually fatal in the absence of drug treatment. HAT infections are caused by the two subspecies *T. b. gambiense* (g-HAT) and *T. b. rhodesiense (r*-HAT). HAT begins as a bloodstream infection (stage I) and progress to an infection of the central nervous system (CNS, stage II), which results in a parasite infection in the brain, and ultimately, death. Following sustained control efforts, HAT infections remain at a historic low with less than 2000 cases per year reported since 2017. However, the at-risk population is estimated at 55 million people and control efforts require effective drug treatments [2,3].

Fexinidazole is a 5-nitroimidazole compound whose potency against trypanosomatids was demonstrated in vivo and in vitro in 1983 [4]. The compound was not pursued clinically at that time because of broad concerns over the potential host toxicity of nitroaromatic compounds. Nitro groups can be highly reactive possessing polarity, hydrogen bonding, and electron-withdrawing chemical properties [5]. Early concerns about potential host toxicity of anti-trypanosome nitro compounds were mitigated by the discovery that trypanosomatids harbor a bacteria-like type I nitroreductase (NTR), which is required for the bioactivation of nifurtimox (5-nitrofuran) and benznidazole (2-nitroimidazole). The fact that trypanosome NTR bioactivation is distinct from that of mammalian type II NTR increased interest in the utilization of nitroaromatic compounds in the treatment of trypanosome infections [6]. NTR activation of nitroaromatic prodrugs in trypanosomes occurs through sequential two-electron reduction of nitro groups to produce amines via nitroso and hydroxylamine intermediates [7]. Incubation of trypanosome NTR with nifurtimox resulted in the formation of bioreactive unsaturated and saturated open-chain nitriles [8]. Whereas benznidazole bioactivation by trypanosome NTR formed a dihydro-dihydroxyl imidazole that

exists in equilibrium with glyoxal and guanidine products [9]. Glyoxal is highly toxic and capable of modifying biomolecules including nucleotides. The trypanocidal consequences of nifurtimox and benznidazole bioactivation are predicted to arise from the formation of toxic levels of DNA damage [8], which has been demonstrated previously for benznidazole [10]. The specific chemical outcomes of NTR activation of fexinidazole have not been reported.

Due to an urgent need for a new treatment for second stage (CNS) HAT, to replace the highly toxic drug melarsoprol, fexinidazole followed an unusual path to clinical usage. In a collaboration between Sanofi and the Drugs for Neglected Diseases initiative (*DNDi*), fexinidazole was proven safe and efficacious for the oral treatment of g-HAT within 10 years of clinical trials [11]. Because of this pathway to approval, little basic research was published on fexinidazole resulting in a considerable knowledge gap regarding the drug's mechanism of action and trypanocidal effects. What is known is that resistance to fexinidazole can arise rapidly in vitro and resistant clones can display cross-resistance against other nitroaromatic compounds, nifurtimox and benznidazole specifically, which may arise via NTR mutations [12,13]. While previous studies have interrogated the effects of nifurtimox and benznidazole on specific drug induced phenotypes [14,10], similar studies have not been published for fexinidazole.

Here we investigated the trypanocidal outcomes of fexinidazole in comparison with nifurtimox and benznidazole while evaluating the proposal that nitroaromatic drugs kill trypanosomes through ROS activation of DNA damage. Specifically, in vitro assays of the model African trypanosome *T. b. brucei*, a livestock pathogen, were interrogated to determine the effects of nitroaromatic drugs on cell cycle, DNA synthesis, ROS stress, and DNA damage formation. Our findings show that, while all three drugs can result in ROS production, fexinidazole treatment results in a profound and inhibition of DNA synthesis. In addition, we developed a flow cytometry assay of DNA damage based on γH2A-phosphorylation to monitor the formation of DNA breaks over cell cycle stages. Data generated from the γH2A-phosphorylation show that fexinidazole treatment can result in significant levels of DNA damage in S phase and $G_2$ after as little as 3 hours of treatment, suggesting that DNA damage formation may be a key factor in fexinidazole's trypanocidal activity. The putative mechanistic implications of fexinidazole's cytotoxic effects will be discussed. These findings also highlight the importance of anti-trypanosome drugs that inhibit DNA synthesis.

## Methods

### Culture and drug treatments of *T. brucei*

Bloodstream form *T. brucei* (*Lister427*) single marker (SM) parasites were used in all untreated and drug treated experiments herein [15], which were conducted in HMI-9 media in vitro [16]. Supplier provided anti-trypanosome drugs, benznidazole (Sigma-Aldrich, 419656), fexinidazole (SelleckChem, S2600), and nifurtimox (Sigma- Aldrich, N3415), as powders that were resuspended in their designated solvent and stored at -20°C prior to use. For experimental use each drug was diluted in HMI-9 to their final concentration as shown for each experiment.

Drug efficacy and parasite death during drug treatments were evaluated in three ways: 1) cumulative growth assays, 2) death flask assays, 3) $EC_{50}$ determination by AlamarBlue (ThermoFisher, DAL1100) cell viability assay.

1) Cumulative growth assays were conducted in 12 well culture dishes in 2 mL of HMI-9 or HMI-9 with drug added. Parasites were seeded at 100,000 cell/mL, parasite growth recorded after 24 hours and diluted back to 100,000 cell/mL every day for a period of 1 week.

2) "Death flask" assays were utilized to observe the timing of parasite death over 1–2 weeks as well as the occurrence of spontaneous drug resistance under drug selection. Tissue culture flasks (T-25) were inoculated with 10,000 cells/mL in 5–10 mL of HMI-9 and counted daily by hemocytometry. Every 3 days, cultures were centrifuged (1,500 RPM, 10 minutes) to isolate parasites and transferred to a new flask containing fresh media and drug in the appropriate drug treatment condition.

Daily counts from growth and death experiments were recorded and graphed using GraphPad Prism. All experiments included three biological replicates and error bars represent the mean and standard deviation, raw hemocytometry counts can be found in S1 Data.

3) $EC_{50}$ determination was conducted in 96 well plate assays using AlamarBlue to determine cell viability during drug treatments. Parasites were plated at 25,000 cells/ well and treated for 48 hours prior to the addition of 20 μL of AlamarBlue per well and 4 hours of incubation (manufacturers protocol optimized for *T. brucei*). Plates were then read on a SpectraMax i3x Multi-Mode Microplate Detection Platform at 590 nm emission and 530 nm excitation fluorescence wavelength. AlamarBlue fluorescence intensity was converted to percent (%) death using puromycin treated controls as 100% death based on the following formula: % dead = 1-(avg test abs – avg puro abs)/ (avg UT abs– avg puro abs). The resulting data over a range of drug concentrations was then graphed to show % death vs. log of the nM concentration of drug using GraphPad Prism for at least 3 biological replicates. Note* average $EC_{50}$ determination for benznidazole, fexinidazole, and nifurtimox shown in S1 Fig.

## Cell cycle analysis by flow cytometry and associated statistical analysis

Standard flow cytometry approaches were used to measure cell cycle progression using propidium iodide staining of approximately 5 million cells following formaldehyde fixation as described [17]. Cell cycle analysis and all flow cytometry analyses performed herein, were performed using a BDFACS Celesta and FACSDiva software for acquisition analysis. The resulting data were then analyzed and visualized using the FlowJo analysis package. Cell cycle gating (AN, $G_1$, S phase, $G_2$, and multinucleated) was generated in FlowJo for at least 3 biological replicates for each treatment condition then underwent statistical analysis using GraphPad Prism with multiple t-tests and unpaired with parametric test.

## Immunofluorescence microscopy

*T. brucei* parasites were grown under shown conditions (UT or drug treated) in a manner that enabled the isolation of 5–10 million cells per condition. Parasites were isolated, resuspended in PBS and 1% formaldehyde and allowed to settle on poly-L-Lysine coated slides (Electron Microscopy Service, 63410–01). Parasites were then stained with an anti-VSG-2 monoclonal antibody [18] conjugated to Alexa647 and mounted in Vectashield (Vector Laboratories, H-1800–10) containing 4′,6-diamidino-2-phenylindole (DAPI). Slides were visualized on a Leica DMi8 Inverted Fluorescent Microscope and images were analyzed using the Fiji (ImageJ) software package for the identification and counting of the number of nuclei (N) and kinetoplasts (K) per parasite for at least 100 parasite cells per condition. Final image preparation was completed in Adobe Photoshop in keeping with standard guidelines.

## DNA synthesis quantification by BrdU incorporation assay

DNA synthesis can be evaluated in context with cell cycle to determine the timing of cell cycle associated defects in DNA synthesis [19,20]. Exponentially growing *T. brucei* was prepared for BrdU incorporation assays similar to previously described assays [21,22]. Parasites were incubated with 100 μM of 5-bromo-2-Deoxyuridine (BrdU, Sigma Aldrich B5002) for 1 hour at 37°C, harvested by centrifugation (1,500 RPM), washed 2x in PBS (phosphate-buffered saline), and fixed with 1% paraformaldehyde for 20 minutes. Fixed cells were then permeabilized with 0.1% Triton X-100 for 30 minutes, treated with 3 M HCl, washed 3 times in PBS, and incubated with anti-BrdU antibody conjugated with Alexa Fluor 647 (1:250 dilution, ThermoFisher B35140), 1:1000 DAPI, and 0.5% BSA overnight. Prepared cells were then washed with PBS and resuspended for flow cytometry analysis as described.

## H2DCFDA quantification of ROS in *T. brucei*

Parasites were seeded at 100,000 cells/ mL and treated with drug or control for 24 hours prior to H2DCFDA assays. After 24 hours in the presence of drug or control condition, parasites were pelleted and washed twice in PBS supplemented

with glucose (20 mM) prior to resuspension PBS + glucose (20 mM). Cells were then incubated with H2DCFDA (20 μM, ThermoFisher D399) for 30 minutes at 37°C and then washed twice with PBS+glucose prior to staining with SYTOX Orange (5 nM, ThermoFisher S11368) at room temperature for 15 mins prior to flow cytometry analysis. Dead cells (SYTOX positive population) were gated out of total cells, to result in Live Cell population that was the basis of histograms used in ROS analysis based on H2DCFDA fluorescence (A488 channel).

### Flow cytometry analysis of DNA damage using anti-γ-H2A-Phos antibody

Anti-γH2A-Phos antisera were raised in rabbits using the KLH-conjugated phosphor-peptide, C-KHAKA[pT]PSV as described previously [23] using ThermoFisher Scientific custom peptide synthesis and antibody services. The resulting anti-sera were purified and conjugated to Alexa488 using standard protocols. Validation of anti-γH2A-Phos detection of DNA damage in *T. brucei* was validated inhouse by western blot analysis (S8 Fig) and immunofluorescence microscopy of DNA break foci prior to adaptation of the antibody to flow cytometry-based analysis of DNA damage herein. To evaluate DNA damage, as marked by γH2A histone phosphorylation, by flow cytometry *T. brucei* parasites were grown in the indicated UT or drug treated condition harvested by centrifugation and washed in cold PBS. Cells were then fixed in 1% paraformaldehyde and permeabilized in 0.1% Triton-X prior to staining with anti-γH2A-Phos-A488 used at a 1:100 dilution and DAPI. All samples were evaluated by flow cytometry to identify the cell gate, singlets, and the A488 populations associated with unstained, anti-γH2A-Phos-A488 stained, and DNA damaged (γH2A-P+) populations, positive control was phleomycin (PH, 5 μg/mL) treated for 3 hours (S7 Fig). Gating is shown as an anti-γH2A-Phos-A488 vs. DAPI to visualize the formation of DNA damage over stages of cell cycle.

### Statistical analysis of flow cytometry data

Flow cytometry data were collected on a BD Celesta Cell Analyzer using BDFACSDiva Software. The raw exported flow cytometry (.fcs) files were analyzed using Flowjo Single Cell Analysis Software version 10.10.0. Flowjo gating strategies were presented as supplemental data associated with each flow cytometry-based figure. Raw counts were exported, along with the number of replicates used in each experiment, to generate supplemental data table for each figure. The relevant outcomes, final counts and percentages, were exported to Graphpad Prism (Version 10.5.0) for graphical and statistical analyses. All statistical analyses of flow cytometry data were based on at least 3 biologically independent replicates where error bars represent the mean and standard deviation from the mean. Statistical significance testing is presented as Pvalues based on Unpaired Students T-tests using a two-tailed analysis with a Pvalue less than 0.05 considered as significant. Additional levels of significance and associated symbols can be found in the figure legends and associated supplements.

## Results

### Establishing trypanocidal dynamics of nitroaromatic drug treatments

This study seeks to determine the specific effects of nifurtimox (NIF), benznidazole (BENZ), and fexinidazole (FEX) on trypanosome cytotoxicity in the context of the predicted cellular effects of nitro-drugs, namely ROS stress and DNA damage. The $EC_{50}$ of each drug has been reported as NIF 7μM, BENZ 35 μM, and FEX 3 μM, which was supported in our own cell viability assays (S1 Fig). Previous studies comparing anti-trypanosome drug cytology used a consistent multiplicity of 5x the drug $EC_{50}$ [14]. In contrast to this approach, we aimed to identify drug concentrations that resulted in parasite death over a similar timeframe and with similar kinetics. In addition, we sought to identify drug concentrations with sub-lethal effects during the first 24–48 hours of drug treatment to enable meaningful biological comparisons of drug effects prior to parasite death. Thus, trypanosome killing was evaluated in two ways: A) cumulative growth assays (Fig 1A) that involve the daily passage of parasites and B) death assays (Fig 1B and 1C), in which parasites are permitted to die in the presence of drug without cell passage. To ensure consistent nutrient supply and drug concentration, death assays included

centrifugation and media replacement with fresh drug added every 3 days. The death assays also enabled us to observe the occurrence of spontaneous drug resistance to nifurtimox and fexinidazole (Fig 1B), which had been reported previously [12,13].

In cumulative growth assays we found that NIF 6 µM (~1x $EC_{50}$), BENZ 60 µM (~2x $EC_{50}$), and FEX 20 µM (6x $EC_{50}$) all resulted in a significant decrease in parasite growth without eliminating the parasites in 4 days (Fig 1A). In contrast, treatment with 12 µM NIF (~2x $EC_{50}$) and 120 µM BENZ (~3x $EC_{50}$) resulted in complete absence of countable parasites after 72 hours. Treatment with 50 µM FEX resulted in no countable parasites after 48 hours in cumulative growth assays. These data were then used to establish LOW and HIGH drug concentrations in parasite death assays that retained parasite viability between 0–48 hours. We identified concentrations for each drug that resulted in the onset of parasite death (reduced cell viability) between day 2 and 4, with the limit of detection (10,000 cells/mL) being reached

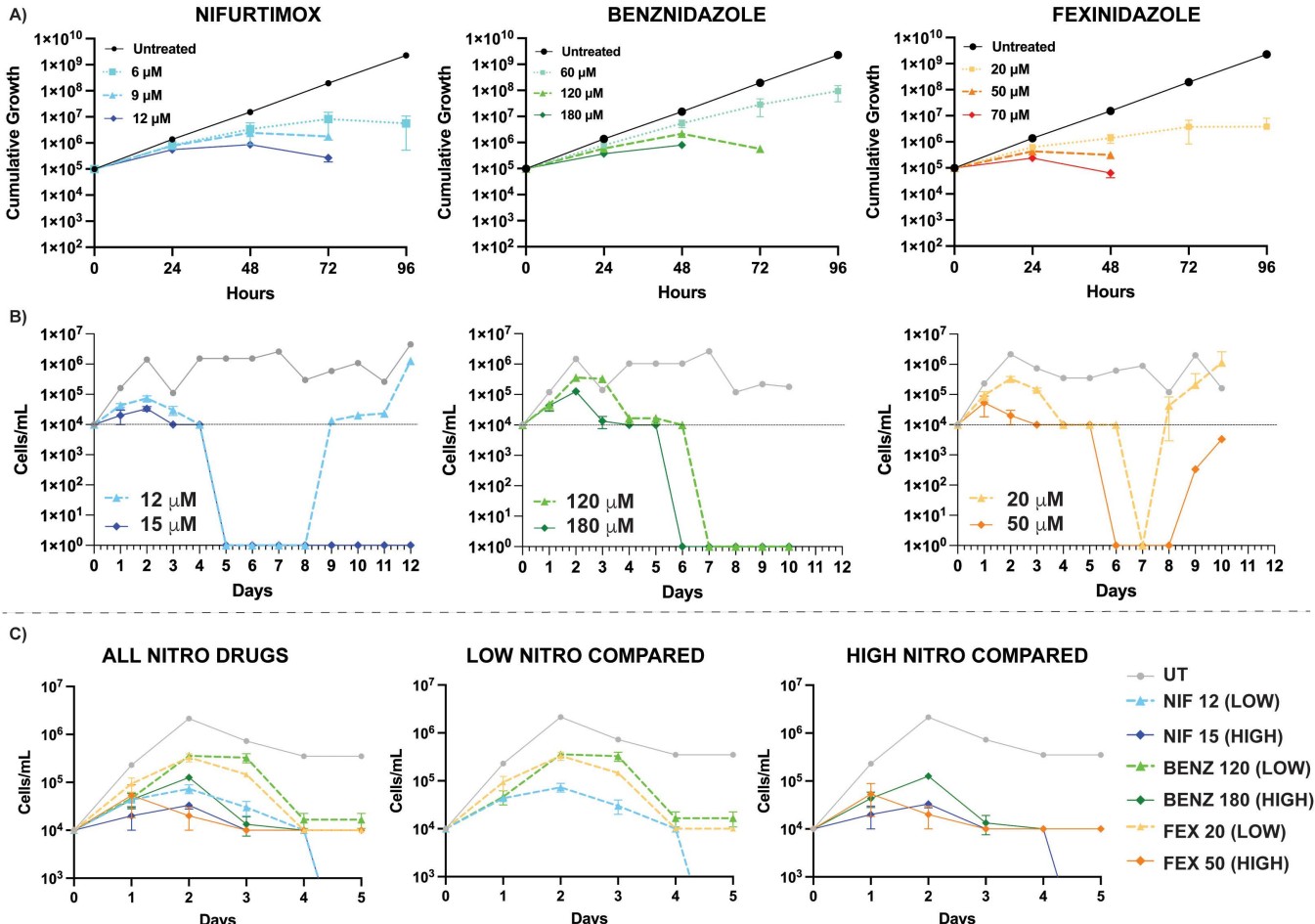

**Fig 1. Trypanocidal activity of nitroaromatic drugs.** A) Cumulative Growth Assays were conducted during treatment with nifurtimox (Blue), benznidazole (Green), and fexinidazole (Orange) using concentrations shown in figure legends. Daily dilution of parasites to 100,000 cells/mL and quantification of their cumulative growth over 5 days. The termination of a line before 96 hours indicates parasites that were too few to pass. B) *T. brucei* death was monitored for 10-12 days during treatment in Nifurtimox (Blue) at 12 µM (LOW) & 15 µM (HIGH), Benznidazole (Green) at 120 µM (LOW) & 180 µM (HIGH), and Fexinidazole (Blue) at 20 µM (LOW) & 50 µM (HIGH). C) More detailed visualization of data shown in panel (B) for parasite counts ($10^3$-$10^7$) from day 0 to day 5. Comparisons of all three drugs at both LOW and HIGH concentrations are graphed. Data shown were derived from 3 independent biological replicates with error bars derived from the mean and standard deviation using GraphPad Prism. Spontaneous resistance arising during NIF 12 µM and FEX 50 µM treatments arose in only one of the three replicate populations, thus no error bars at associated with these lines in panel B.

after day 3 of treatment. A comparison of drug concentration, $EC_{50}$, and treatment outcomes can be found in S1 Table and raw counts from hemocytometry are shown in S1 Data. The LOW drug concentrations (NIF 12 µM, BENZ 120 µM, & FEX 20 µM) sustained some level of parasite growth for at least 48 hours. All LOW drug concentrations resulted in parasites at or below the limit of detection by day 4 (Fig 1B). The HIGH drug concentrations (NIF 15 µM, BENZ 180 µM, & FEX 50 µM) permitted growth from 0 to 24 hours in all drugs, with NIF & BENZ displaying growth from 24-48 hours. All HIGH drug treatments resulting in parasites at or below the limit of detection by day 3 (Fig 1B). While imperfect, the LOW and HIGH drug concentrations empirically established here provided the most comparable drug concentrations of NIF, BENZ, and FEX for evaluating cytology phenotypes between 0 and 48 hours of treatment and will be used throughout this study.

**Fexinidazole induced cell cycle defects differ from nifurtimox and benznidazole**

To begin to understand the cytotoxic effects of fexinidazole on *T. brucei* we evaluated the cell cycle effects at the established LOW and HIGH nitroaromatic drug concentrations (Fig 1B). In untreated (UT) cells, $G_1$ accounts for the majority of the cell population (~60%), $G_2$ is the next largest (~20%), and DNA synthesizing cells in S phase (~15%) is the last major population (Fig 2 – Untreated). Minor populations can also include anucleated (AN) cells that form a sub-$G_1$ peak, less than 2% of the population of healthy cells, and multinucleated (MN) cells (~1%) with a greater DNA content than $G_2$, both of which can arise from cytokinesis defects. Cell cycle analysis was conducted for all nitro-drugs at both LOW and HIGH concentrations following 24 and 48 hours of drug treatment. The cell cycle gating strategies employed in the analysis of these data can be found in S2 Fig.

Treatment with NIF and BENZ resulted in a general trend of a decreased $G_2$ population over increasing time and drug concentration (Fig 2A). Notably, the decrease in $G_2$ cells during NIF treatment resulted in loss of the $G_2$ peak (corresponding with an increase in $G_1$), while BENZ treatment resulted in a change in the shape of $G_2$ that included cells with higher DNA content in the region between 100K & 150K relative units of fluorescence from propidium iodide (PI) DNA content staining. While this is partially gated in the multinucleated region of the plot, the BENZ treated parasites did not form truly multinucleated cells, which would be observed at 200K (representing a doubling of DNA content compared to $G_2$ without cytokinesis). All raw counts, percent populations, and data arising from statistical analyses of cell cycle flow cytometry herein can be found in S2 Data. Treatment with HIGH NIF after 24 hours resulted in a statistically significant decrease in the population of parasites in $G_2$ (Fig 2B). Reduction of $G_2$ populations was statically significant in all NIF and BENZ treatments after 48 hours (Fig 2B, with Pvalues shown in S3 Fig). In contrast, treatment with FEX resulted in the progressive loss of S phase populations over increasing time and concentration. After 48 hours, both LOW & HIGH FEX concentrations resulted in a statistically significant decrease in the S phase population (Fig 2B). In addition, we evaluated the statistical significance of the changes in S phase through statistical comparison of LOW or HIGH FEX concentrations with the cognate LOW or HIGH concentrations of NIF and BENZ at both 24 and 48 hours (Fig 2B, Orange asterisks over error bars). We found that the FEX specific reduction in S phase was statistically significant when compared to NIF, with the difference after 48 hours in HIGH drug having a Pvalue less than 0.0001 (Fig 2B and S2 Data). The FEX induced decrease in S phase was also significant in comparison with BENZ in all but one instance (Fig 2B, BENZ 48 hours). Thus, the effect of fexinidazole on the reduction of parasites in S phase is both significant and unique to fexinidazole among the nitroaromatic drugs compared here (Fig 2B).

To better understand the outcomes of fexinidazole treatment we prepared histogram overlays for drug comparisons (Fig 2C) and over increasing FEX concentrations up to 70 µM treatment over 48 hours (Fig 2D). The highest concentration of fexinidazole did not result in the loss of $G_2$ cells but did display a significant loss of S phase cells and an increase in the sub-$G_1$ anucleated cell population (Fig 2D). In summary, the most significant effect of NIF and BENZ was loss of the $G_2$ population and FEX treatment resulted in a significant reduction in the number of parasites in S phase (S3 Fig demonstrates statistical significance).

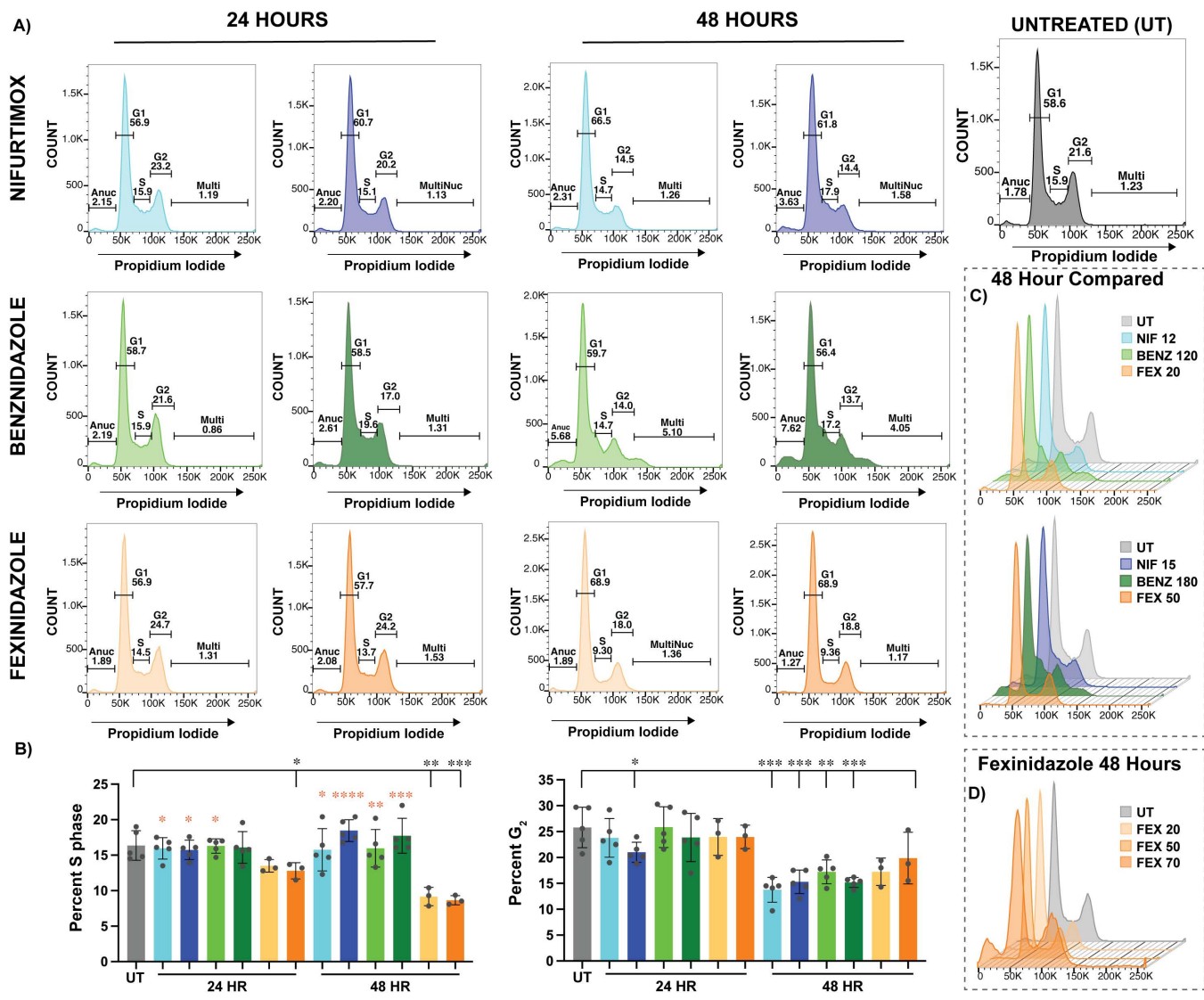

**Fig 2. Cell cycle effects of nitroaromatic drugs.** A) Untreated (UT) cell cycle traces were compared to LOW & HIGH nitroaromatic drug concentrations after 24 hours (Left side) and 48 hours (Right side) of drug treatment. Percent of each cell cycle population, based on UT gating, are shown. Additional information on the cell cycle gating strategy used can be found in S2 Fig. Colors used to denote drug concentration are LOW: NIF 12 µM – light blue, BENZ 120 µM – light green, FEX 20µM – light orange or HIGH: NIF 15 µM - dark blue, BENZ 180 µM – dark green, FEX 50 µM – dark orange. B) Bar graphs depicting the percent of cells in S phase (Left) and $G_2$ (Right) show the effects of drug treatments (UT- Grey, NIF LOW – Light Blue, NIF HIGH Dark Blue, BENZ Low – Green, BENZ HIGH – Dark Green, FEX LOW – Light Orange, FEX HIGH – Orange) after 24 or 48 hours. Statistical significance of drug treatments in comparison with UT are shown on the line above the graph (Black) with all statistically significant events resulting from students t-test as described in the methods with P values *<0.05, **<0.01, ***<0.001, and ****<0.0001 additional details can be found in S2 Data. Additional statistical comparison between LOW or HIGH FEX and the corresponding LOW or HIGH concentration of NIF or BENZ were performed for S phase with statistically significant differences between FEX shown as Orange asterisks. C) Staggard offset cell cycle graphs are used to compare all drugs at LOW (Top) and HIGH (Bottom) concentrations after 48 hours of treatment. D) For further comparison of the outcomes of FEX treatment, an additional concentration of FEX 70 µM was analyzed and visualized using staggard offset cell cycle plots along with established LOW and HIGH concentrations of FEX.

## Fexinidazole treatment decreases S phase parasite population

To evaluate the cell cycle phenotypes of nitroaromatic drugs in greater detail, we analyzed the cellular composition of kinetoplasts and nuclei by immunofluorescence (IF) microscopy using DAPI staining. Parasites in the family kinetoplastida are unique among protists in that they harbor a mitochondrial DNA containing sub-organelle called the kinetoplast, which defines their phylogeny. During the cell cycle, entry into S phase can first be visualized with the elongation and subsequent division of the kinetoplast prior to the completion of nuclear DNA (nDNA) synthesis and division of nuclei [24]. Thus, the trypanosome cell cycle can be monitored based on the number of kinetoplasts and nuclei per cell. Trypanosomes in $G_1$ harbor one kinetoplast and one nucleus (1K1N). Upon S phase entry the kinetoplast elongates and divides resulting in 2K1N parasites. The $G_2$ population is marked by the division of nuclei resulting in 2K2N parasites, which normally proceed to cytokinesis [24,25]. The fact that trypanosomes lack mammalian-like cell cycle checkpoint control means that DNA synthesis can proceed even in the event of damaged DNA [25]. The outcomes of these events can include the inhibition of cytokinesis, resulting in multinucleated cells (4N or greater), or asymmetric cytokinesis events that result the formation of an anucleated daughter (0N) with the parent retaining 2N, which can appear as a $G_2$ cell cycle stall.

Cell cycle progression was analyzed using IF microscopy of cells that were stained with an anti-VSG-2 coat antibody to visualize the cell surface and DAPI to visualize kinetoplasts and nuclei (Fig 3). At least 100 cells were analyzed from each drug treatment condition, for which the number of kinetoplasts and nuclei were counted for each cell. Aberrant cell types were annotated as other, and an example can be seen in S4 Fig. Analysis of IF microscopy for UT parasites resulted in an average of 58% 1K1N ($G_1$), 27% 2K1N (S phase), 12% 2K2N ($G_2$), and 2% other (S4 Fig, shows all percentages from each population from data in Fig 3). In general, treatment with NIF or BENZ resulted in a decrease in the 2K2N populations corresponding to $G_2$, resulting in as little as 8% (NIF HIGH 24 hours) and 7% (BENZ HIGH 48 hours) 2K2N (Fig 3B). Despite the overall pattern of NIF and BENZ decreasing the 2K2N parasites, there were instances where 2K2N increased, NIF HIGH 48 hours 14% and BENZ LOW 24 hours 14%. In contrast, FEX treatment resulted in a marked decrease in 2K1N cells associated with S phase and an increase in 2K2N cells associated with $G_2$ (Fig 3B). It is noteworthy that the increase in 2K2N cells by this method differs somewhat with the FEX flow cytometry data presented in Fig 2, where there was no statistically significant difference in $G_2$ populations. However, we did observe by flow cytometry that at very high concentration of FEX (70 µM) there is an accumulation of parasites in $G_2$. Thus, this difference between analytical methods (microscopy vs. flow cytometry) highlights the importance of evaluating these outcomes using multiple methods. Nonetheless, the observed decrease in 2K1N cells during fexinidazole treatment correlates with the decrease in the S phase population visualized by flow cytometry and suggested that fexinidazole treatment may result in a DNA synthesis defect.

## Fexinidazole treatment results in a DNA synthesis defect

To further characterize the observed decrease of S phase populations during fexinidazole treatment, DNA synthesis was measured during treatment with the three anti-trypanosome nitroaromatic drugs (Fig 4). We employed a flow cytometry-based bromodeoxyuridine (BrdU) incorporation assay, in which parasites were pulsed with the alternative base BrdU and then DNA synthesis is measured using an anti-BrdU antibody to determine the proportion of BrdU positive, DNA synthesizing, cells [21,22]. Cells were also stained with DAPI to identify the cell cycle specific populations of cells with BrdU incorporated. Quantification of DNA synthesis was measured after 6, 12, or 24 hours of drug treatment to observe the effects leading up to the cell cycle effects reported in Figs 2 and 3. The proportion of the population undergoing DNA synthesis was gated based on the anti-BrdU (Alexa647) signal, in which untreated parasites resulted in 38–48% of the parasites undergoing DNA synthesis over 6–24 hours (Fig 4). The gating strategy use in this analysis can be found in S5 Fig with raw flow cytometry counts and statistical analyses shown in S3 Data. HIGH concentrations of NIF, BENZ, and FEX were analyzed at 6, 12, and 24 hours over three biological replicates for each treatment condition and timepoint. After

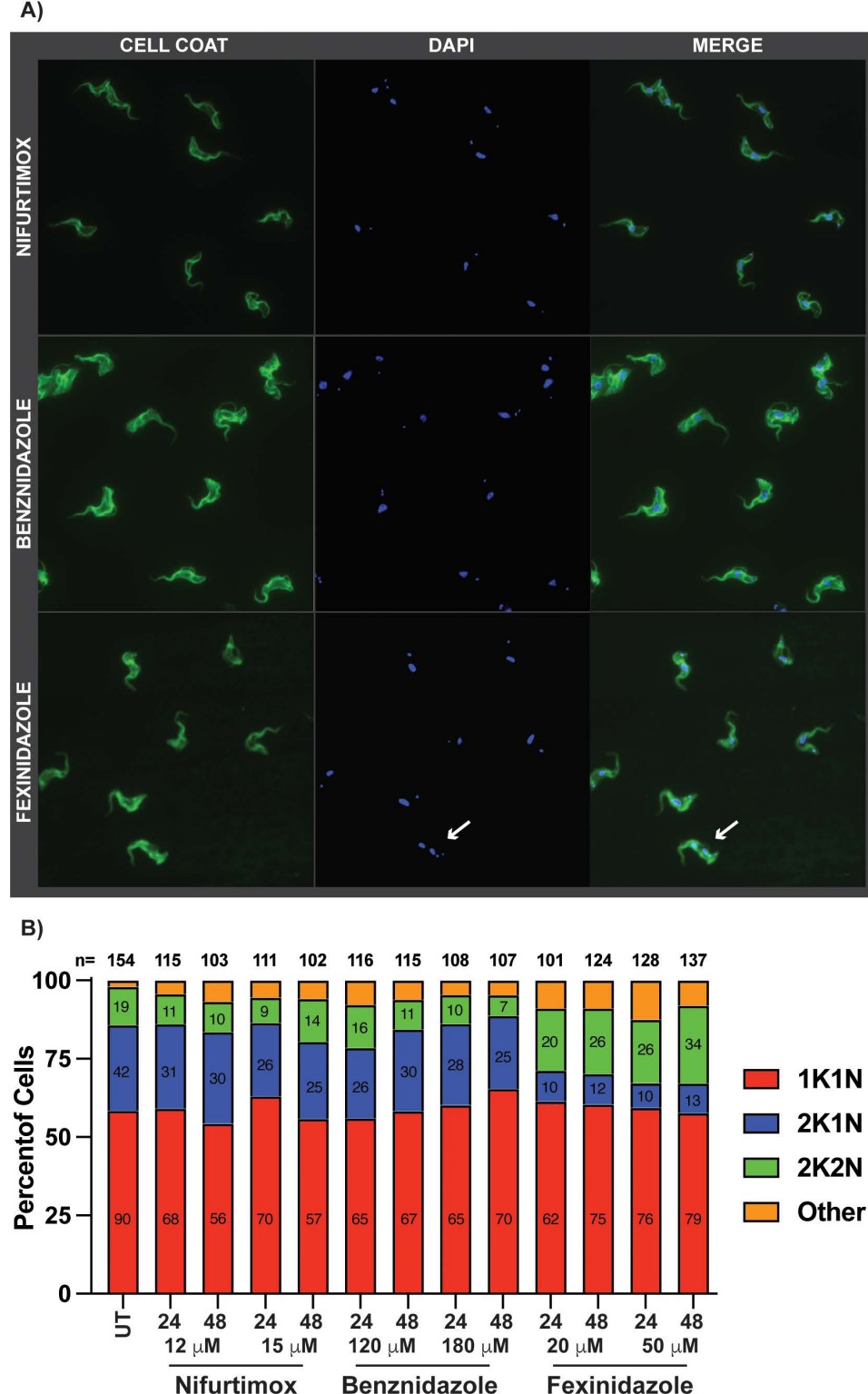

**Fig 3. Nitroaromatic drug effects on parasite kinetoplast (K) and nuclei (N) composition.** A) Sample IF microscopy images for *T. brucei* treated with NIF, BENZ, and FEX after 24 or 48 in drug concentrations shown. UT shown in S4 Fig. Graph generated from IF cell counts of nitroaromatic drug treated parasites after 24 and 48 hours in HIGH and LOW drug concentrations. At least 100 parasites were counted per condition with the total counted

parasites shown as n = value at the top each bar on the graph. Analysis of kinetoplast (K) and nuclei (N) counts per cell analyzed in FIJI with 1N1K (red), 2N1K (blue), 2N2K (green & arrow), and patterns outside these parameters shown as Other (orange). Additional FEX induced cell cycle defects can also be found in S4 Fig.

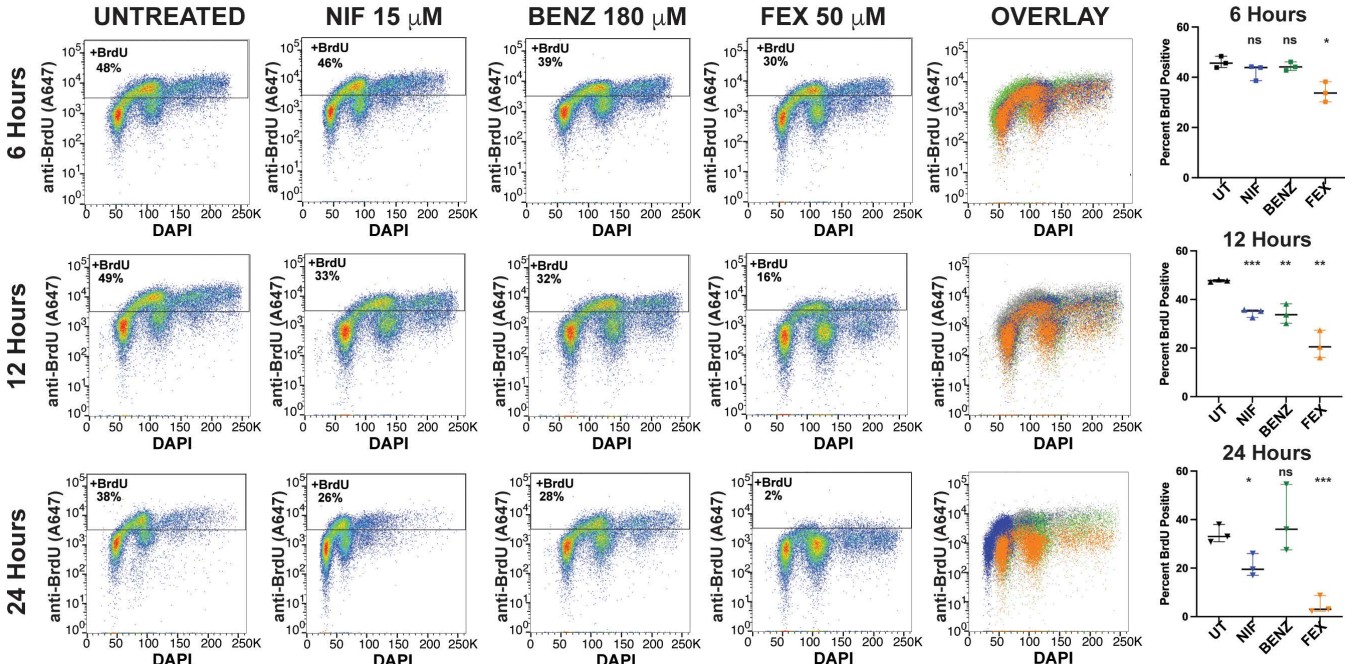

**Fig 4. Analysis of DNA synthesis during nitroaromatic drug treatment by BrdU incorporation.** Untreated or nitroaromatic drug treated *T. brucei* was pulsed during in vitro growth with 100 µM BrdU for 1 hour before parasites were harvested for flow cytometry analysis [22]. For each condition, the DNA synthesizing population was gated based on anti-BrdU-A647 antibody staining and compared with UT at 6 hours, 12 hours, and 24 hours post treatment alongside DAPI (x-axis) DNA content staining for cell cycle. The effects the HIGH concentrations of NIF (15 µM), BENZ (180 µM), and FEX (50 µM) are shown at each time point and overlay. Graphical analysis of biological triplicates is shown at each timepoint for UT and each drug treatment along with statistical significance, Pvalues * < 0.01, ** < 0.001, *** < 0.0005. Additional statistical information and raw counts can be found in S6 Fig.

6 hours of treatment neither NIF nor BENZ had a significant effect on DNA synthesis, whereas FEX had a modest but statistically significant reduction in DNA synthesis (Fig 4). After 12 hours all three drugs resulted in a statistically significant decrease in the DNA synthesizing cell population. However, FEX averaged 20% DNA synthesizing cell population, which was almost a 50% reduction when compared with UT (48%) parasites at the same time point. In contrast, NIF and BENZ retained more than 30% of their DNA synthesizing cell population. (Fig 4). After 24 hours of FEX treatment nearly the entire DNA synthesizing population was lost (2% remaining) in comparison with a minor reduction during NIF treatment (26%) and BENZ treatment having inconsistent results that lacked statistical significance. Thus, while all three drugs can impact DNA synthesis, FEX had a specific and pronounced effect resulting in near cessation of DNA synthesis within 24 hours of treatment (Fig 4). These data are in agreement with cell cycle analyses indicating that FEX treatment results in a decrease in S phase parasite populations.

## Induction of ROS stress by nitroaromatic drugs

Despite the long-standing prediction of ROS formation as a cytotoxic outcome of nitro-drug treatment this has not been well investigated in trypanosomes [26]. To measure ROS in living parasites during drug treatment we

employed the cell-permeant dye 2',7'-dichlorodihydrofluorescein diacetate (H2DCFDA), which converts to fluorescent 2',7'-dichlorofluorescein (DCF) upon cleavage by intracellular esterases or oxidation [27]. Flow cytometry-based assays were conducted using SYTOX Red [28] to gate out the dead cell population and H2DCFDA fluorescence (488 nM) as an indicator of ROS induction. H2DCFDA stained parasites were established with a relative fluorescence units (RFU) value between $10^2$ and $10^3$, which then shifted to greater than $10^3$ when treated with $H_2O_2$ (75μM, positive control) for 24 hours. Thus, a ROS positive gate was established for cells with a H2DCFDA RF > $10^3$ (Fig 5A), which for $H_2O_2$ treatment resulted in 20–65% of the live parasites being positive for ROS stress. The gating strategy used in this analysis can be found in S6 Fig with raw flow cytometry counts and statistical analyses shown in S4 Data. Initial experiments conducted in an effort to observe ROS effects at less than 24 hours did not result in any measurable H2DCFDA fluorescence shift. Similarly, experiments conducted after 48 hours of $H_2O_2$ resulted in analysis challenges from poor sample quality. Thus, a duration of treatment of 24 hours was the focus of these analyses.

All nitroaromatic drug treatments at both HIGH and LOW concentrations resulted in a positive ROS shift, though the range of increased H2DCFDA fluorescence values was variable. LOW NIF ranged from 5-22%, LOW BENZ 9–32%, and LOW FEX resulted in 7–39% of the population of cells shifted. Both NIF and FEX resulted in a considerable increase in ROS+ cells at HIGH concentration, NIF 16–42% & FEX 20–52%, while BENZ HIGH surprisingly and consistently decreased at HIGH concentration to an average of 7.7%. All ROS+ shifted populations following $H_2O_2$ or drug treatment were statistically significant in comparison with untreated cells (Fig 5B, Pvalue < 0.0001). Nonetheless, the magnitude of the ROS effect and the separation between UT and treated H2DCFDA florescence peaks was the most pronounced during FEX treatments (Fig 5B) with FEX HIGH in the same range with $H_2O_2$, such that there is no statistically significant difference between them ($H_2O_2$ 75 μM vs. FEX 50 μM, Pvalue = 0.58). Based on these data we concluded that all nitro-drugs tested have the capacity to induce ROS stress after 24 hours of treatment with fexinidazole demonstrating the most pronounced nitro-drug induced ROS producing phenotype.

### Fexinidazole induced DNA damage during S phase and G2

When nitro drugs are activated by the trypanosome type I NTR enzyme, their active forms are predicted to cause a myriad of cytotoxic outcomes that may include DNA damage. Specific DNA damage effects have been previously reported for benznidazole [10]. Here we sought to obtain a more nuanced understanding of DNA break formation within the framework of cell cycle progression. Toward this goal we developed a flow cytometry-based assay for *T. brucei* using a trypanosome specific anti-γH2A-Phos-A488 conjugated antibody [23] that was evaluated alongside DAPI staining to monitor cell cycle. To establish the γH2A-P-based flow cytometry assay, we compared UT cells with phleomycin (PH) treated cells for 3 hours, a drug known to induce DNA break formation. Treatment with PH for 3 hours resulted in 75.4% γH2A-P+ (Alexa488) population in comparison with 2.2% in UT cells (Fig 6). Moreover, PH treatment displayed DNA damage in all cell cycle stages, indicating that DNA damage can be measured in $G_1$, S phase, $G_2$ and MN populations by this method (Fig 6). Based on the clarity of the controls in this newly developed assay we proceeded to evaluate the effects of nitroaromatic drug treatment on the formation of DNA damage.

DNA damage formation was analyzed in HIGH NIF, BENZ, and FEX after 3, 6, and 12 hours of drug treatment to evaluate if observable effects could precipitate the cell cycle defects reported herein (Fig 6A). Gating of the percent of γH2A-P+ (Alexa488) cells in flow cytometry-based assays was based on the fluorescence shift of PH positive control (Fig 6B). The gating strategy used in this analysis can be found in S7 Fig with raw flow cytometry counts and statistical analyses shown in S5 Data. All three drugs (NIF, BENZ, & FEX) resulted in increased DNA damage (γH2A-P+) at all timepoints, when compared to UT control. The percent increase in γH2A-P+ cells in comparison with the total cell population was statistically significant in NIF and BENZ after 12 hours of treatment (Fig 6C, *Pvalue <0.05). In contrast, FEX treatment resulted in statistically significant increases in DNA damage at all timepoints with a greater magnitude of effect than during other treatments (Fig 6C).

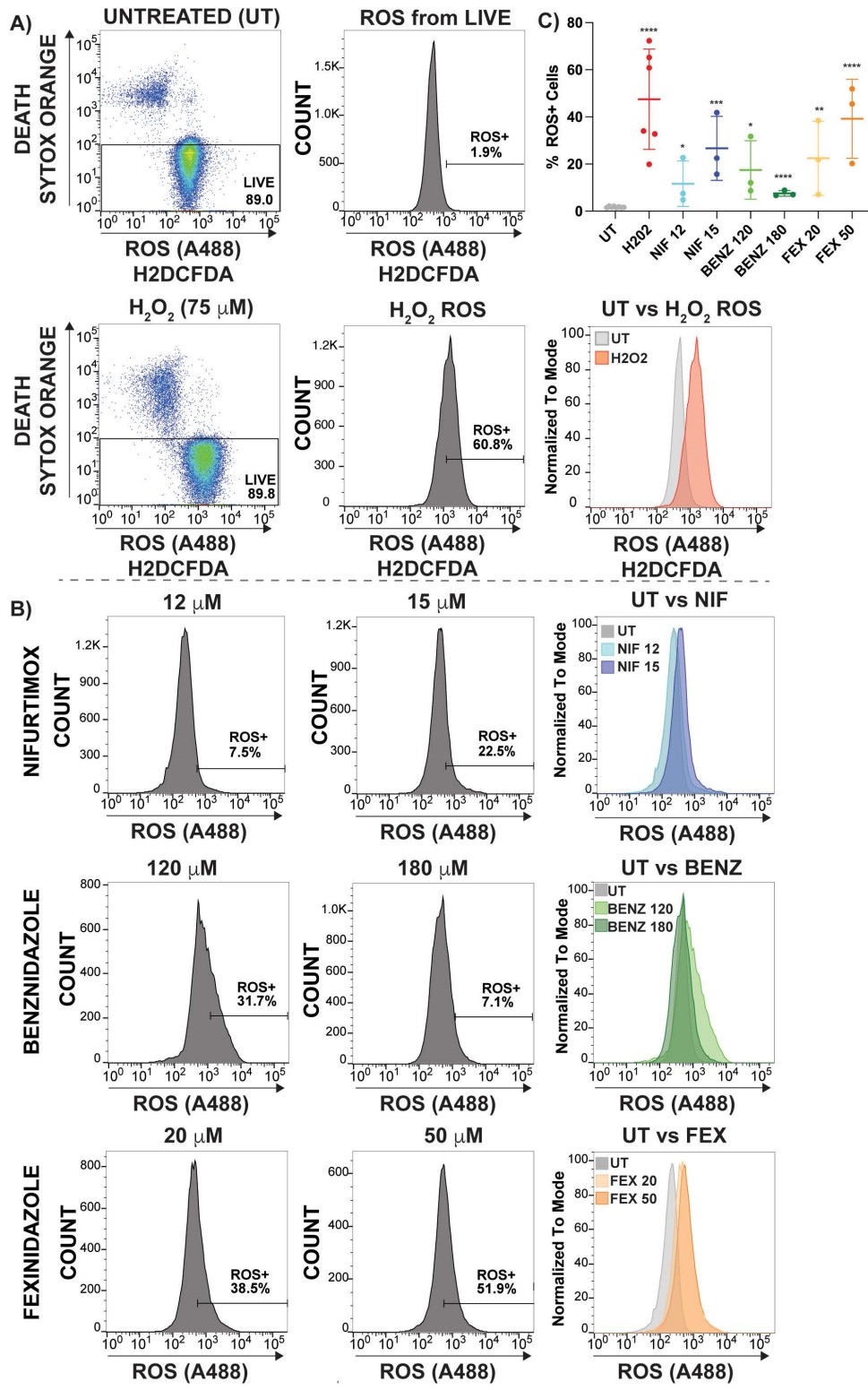

**Fig 5. ROS induction by nitroaromatic drugs, reported by H2DCFDA.** *T. brucei* cells were treated with $H_2O_2$ (Red, 75 µM) or drug at concentrations shown for 24 hours prior to incubation with H2DCFDA (20 µM) and SYTOX Orange (5 nM) staining for live vs. dead cell discrimination. A) Depicts gating strategy with SYTOX Orange on the y-axis gated at greater than $10^2$ relative fluorescence units (RFU) and H2DCFDA staining (Alexa488 channel), with stained cell population between $10^2$-$10^3$ RFU. Upon $H_2O_2$ induced ROS stress the H2DCFDA positive population was shifted to $10^3$ RFU or greater.

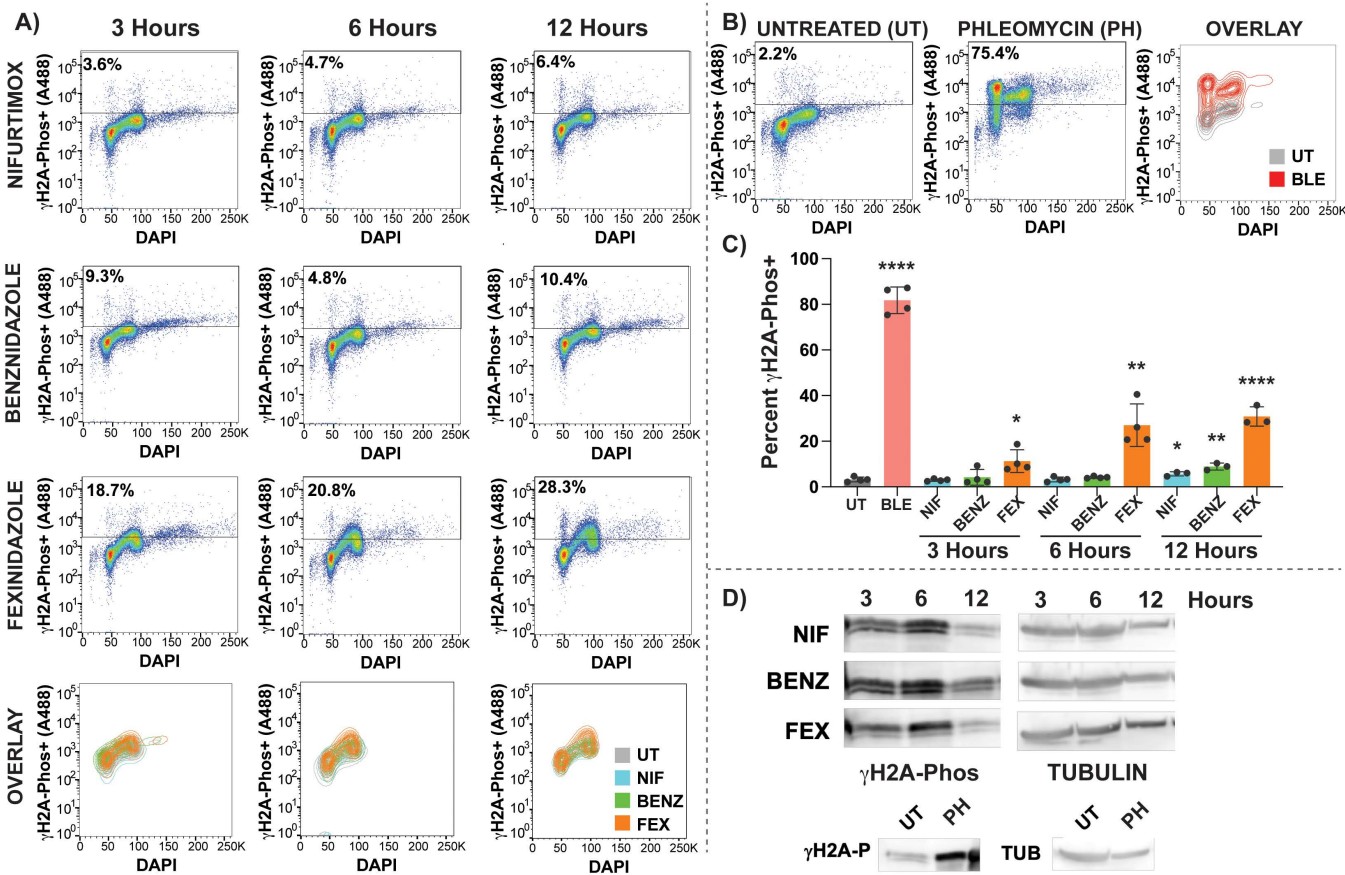

Histograms have gated out Dead (SYTOX Orange positive) parasites. B) Histograms of NIF, BENZ, and FEX are shown for both LOW and HIGH drug concentrations (left and right), including overlay with untreated (UT) H2DCFDA stained cells (grey). The percent ROS+ population is shown for each. C) The ROS+ percent of each population is shown for 3 biological replicates of each drug and concentration along with numerous replicates of UT and H₂O₂ alongside statistical significance of all data in comparison with UT, Pvalues *<0.01, **<0.001, ***<0.0005, ****<0.0001.

**Fig 6. Evaluation of DNA damage during nitroaromatic drug treatment using γH2A-Phos.** An established *T. brucei* specific anti-γH2A-Phos-antibody [23] conjugated to Alexa488 was employed to evaluate the formation of DNA damage alongside DAPI DNA content staining to measure the timing of DNA damage formation during HIGH nitroaromatic drug treatments: A) NIF 15 μM, BENZ 180 μM, and FEX 50 μM after 3, 6, and 12 hours of treatment. B) Gating the γH2A-Phos+ population was established using stained untreated cells (UT) as a negative control and stained phleomycin (PH) treated cells (5 μg/mL, 3 hours) as a positive control for DNA damage formation. The percent γH2A-Phos positive cells from each population were gated using A488 gating shown. C) at least 3 biological replicates from each condition were analyzed graphed and used as the basis for statistical significance, Pvalues based on Unpaired Students T-tests using a two-tailed analysis shown as *<0.01, **<0.001, ***<0.0005, ****<0.0001. D) Western blot analysis using anti-γH2A-Phos-antibody was performed as reported [23] using matched samples in comparison with flow cytometry analysis. Additional replicate data for anti-γH2A-Phos western analysis can be found in S8 Fig.

The anti-γH2A-P+ antibody is commonly used in western blot analysis in *T. brucei* to measure DNA damage. Here, we used matched samples for anti-γH2A-P+ flow cytometry and western blot analysis (Fig 6D). Notably, western blot analysis indicated that all 3 drugs can result in DNA damage at all timepoints (more detail in S8 Fig). Some inconsistency was observed between the magnitude of DNA damage by western blot in comparison with the flow cytometry method (Fig 6C compared with Fig 6D). It is currently unclear why the magnitude of DNA damage is different when comparing western blot and flow cytometry data across timepoints, or which assay is inherently more sensitive. Despite the assay differences,

these data show that all three nitroaromatic drugs were able to induce DNA damage and the anti-γH2A-P+ flow cytometry method provides new insight into DNA damage formation across each stage of cell cycle.

To better understand the effect of fexinidazole on DNA damage formation during cell cycle progression, we quantified the proportion of DNA damaged cells (γH2A-P+) in each stage of cell cycle: $G_1$, S phase, $G_2$, & Multinucleated (Fig 7). The gating strategy use in this analysis can be found in S9 Fig with raw flow cytometry counts, derived percentages, and statistical analyses shown in S6 Data. Untreated cells in $G_1$ and S phase have 2% or less cells in the DNA damage gate, while cells in $G_2$ averaged 3.5%, and multinucleated cell proportions were variable (Fig 7A and S6 Data).

Visualizing the percent of DNA damage over time (3–12 hours) showed that during all nitroaromatic drug treatments $G_1$ cells consistently harbored the least amount of DNA damage while multinucleated cells tend to harbor the most, which is expected for this naturally aberrant cell population (Fig 7B). All drugs tested displayed increasing DNA damage over time in S phase and $G_2$, which were statistically significant in all drugs after 12 hours (Fig 7C). Despite these commonalities, FEX treatment was distinct from the other drugs both in the early onset of DNA damage formation (statistically significant after 3 hours when compared with UT) and the magnitude of DNA damage. After 3 hours FEX treatment resulted an average of 6.5% DNA damage in S phase and 17.8% in $G_2$, which is about 5 times more than either NIF or BENZ. The most pronounced difference was measured after 6 hours where DNA damage arose in an average of 17.5% of S phase cells and 43.7% of $G_2$ treated with FEX compared with 1.4% (S phase) and 3.8% ($G_2$) in NIF and 1.8% (S phase) and 5.1% ($G_2$) in BENZ. Thus, FEX results in 9–12 times the amount of DNA damage observed in the related drugs. Finally, after 12 hours FEX induced 5–9 times the levels of DNA damage as NIF or BENZ during S phase and $G_2$ (Fig 7C, with all data shown in S6 Data). The comparison of FEX induced DNA damage with NIF and BENZ showed that FEX results in statistically higher levels of DNA damage in S phase and $G_2$ than NIF or BENZ at both 6 and 12 hour timepoints (S6 Data).

Since parasites were not cell cycle synchronized in these studies, it is interesting to consider if DNA damage arising in S phase results in cells harboring DNA damage in $G_2$ or if FEX treatment damages DNA in both S phase and $G_2$. One potential implication of these data is that DNA damage arising during FEX treatment may contribute to the FEX induced DNA synthesis defect reported herein (see putative models in Fig 8). Elucidating precisely how fexinidazole treatment results in DNA damage and the interplay between the observed DNA synthesis defect and DNA damage phenotypes reported here will be the subject of future mechanistic studies.

## Discussion

Fexinidazole is positioned to change the landscape of HAT treatment in sub-Saharan Africa by providing patients with an oral therapy against African Sleeping Sickness. However, both current cornerstone therapies against second stage g-HAT, nifurtimox eflornithine combination therapy (NECT) and fexinidazole, now rely on nitroaromatic compounds leaving them vulnerable to drug resistance and cross-resistance [29]. Due to the efficacy of fexinidazole against kinetoplastida parasites, the drug is being actively considered as a possible treatment for American trypanosomiasis and multiple forms of Leishmania infections [30–32]. While fexinidazole's clinical approval may represent a considerable therapeutic advancement, there are major gaps in our understanding of how it kills trypanosomes and potential sources of drug resistance [8,29,33]. This study sought to dissect specific cytotoxic outcomes arising from each of the clinically relevant nitroaromatic drugs: nifurtimox, benznidazole, and fexinidazole. Using cell biology methods, we have observed specific outcomes of fexinidazole that are distinct from related nitroaromatic anti-trypanosomatid drugs, namely DNA synthesis inhibition and the timing and magnitude of drug induced DNA damage formation.

Here we consider the timing and outcomes of trypanosome cytotoxicity during treatment with fexinidazole. The most profound effect, which is likely to promote parasite death, is the near complete inhibition of DNA synthesis over 24 hours of fexinidazole treatment. Nitroaromatic drugs have been implicated in the addition of bulky DNA adducts [34], which can both inhibit DNA synthesis and result in DNA damage formation in other organisms [35]. In fact, inhibition of DNA synthesis by nitroheterocyclic drugs was first documented as early as 1979 [36]. The nature of previously documented

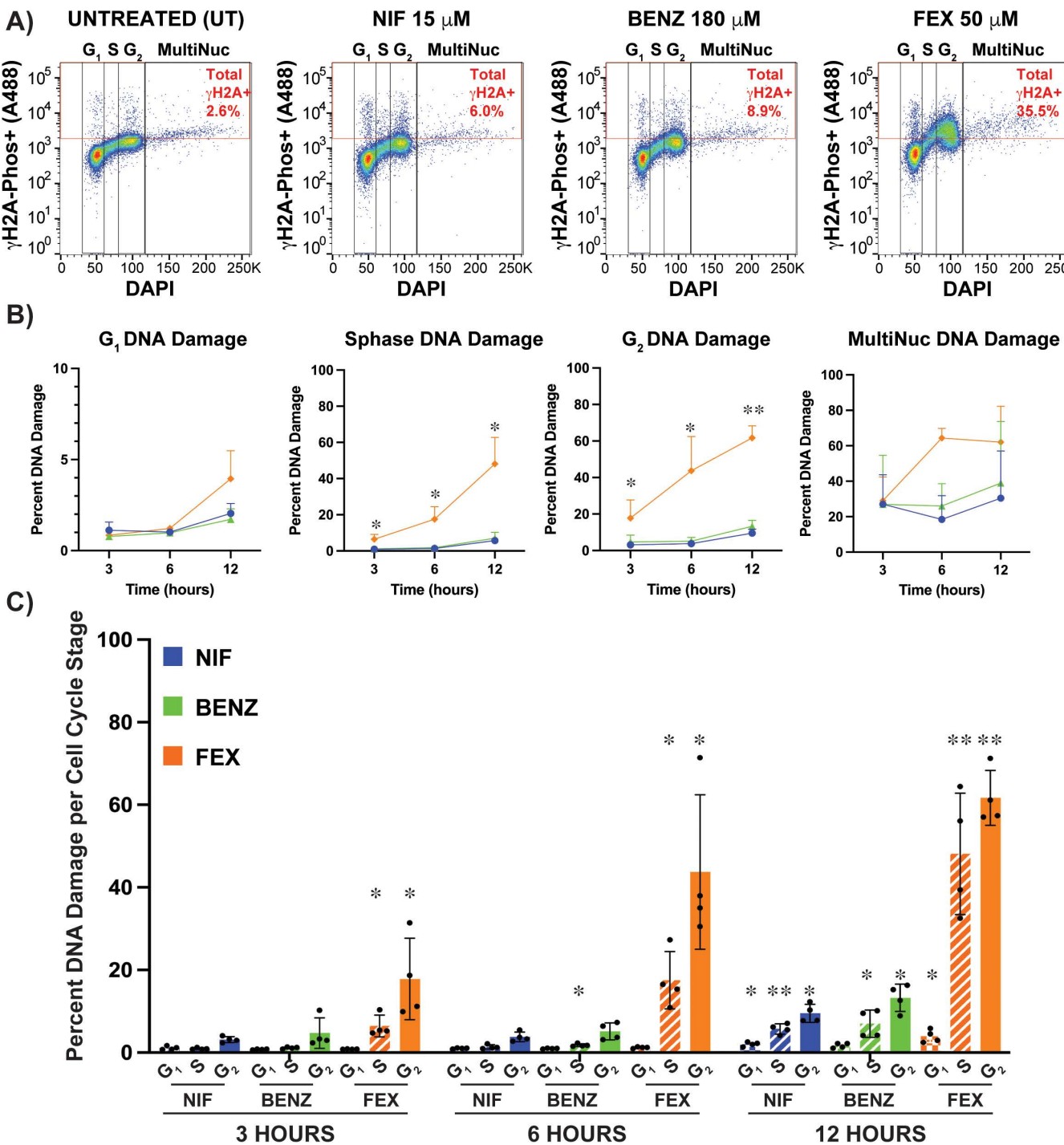

**Fig 7. Analysis of drug induced DNA damage over cell cycle stages.** A) *T. brucei* anti-γH2A-Phos-A488 stained populations were sub-gated based on DAPI staining of $G_1$, S phase, $G_2$, and multinucleated cells. Total γH2A-Phos+ DNA damage shown in red gate and red text for UT, NIF, BENZ, and FEX after 12 hours. Additional gating information can be found in S9 Fig and all raw counts and statistics in S6 Data. S6 Data also includes tables of the number of events (DNA damage cells) in each cell cycle gate shown above. B) The percent of DNA damage arising in each cell cycle stage ($G_1$, S phase, $G_2$, and MultiNuc) was graphed over 3, 6, and 12 hours for NIF 15 μM (blue), BENZ 180μM (green), and FEX 50 μM (orange). Statistically significant differences between FEX and other drugs at each timepoint are shown. C) The percent of DNA damage (γH2A-P+) per cell cycle stage is graphed for NIF (blue), BENZ (green), and FEX (orange) over 3, 6, and 12 hours for $G_1$ (checkered pattern), S phase (stripped pattern), and $G_2$ (solid, no pattern). Statistical significance based on comparison with UT are shown as Pvalues based on Unpaired Students T-tests using a two-tailed analysis shown based four biological replicates with * Pval<0.05, ** Pval<0.005, and ***Pval<0.0005.

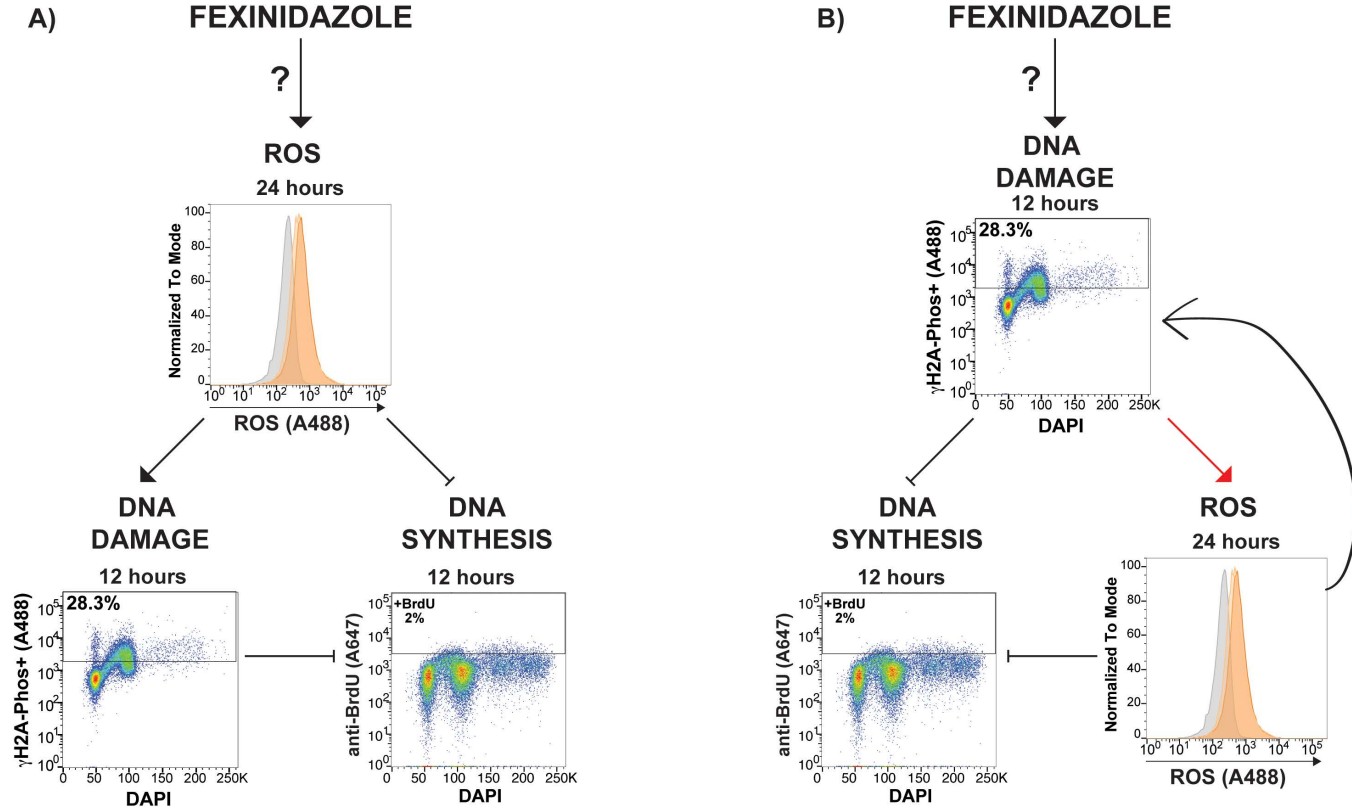

**Fig 8. Contrasting putative models of fexinidazole induced cytotoxicity.** In Model A – Fexinidazole induces ROS stress, by an unknown molecular mechanism denoted with "**?**", to result in DNA damage formation and DNA synthesis inhibition. In Model B – Fexinidazole results in DNA damage formation, by an unknown molecular mechanism denoted with "**?**", that results in DNA synthesis inhibition and ROS formation (red arrow line indicates currently unknown pathway in *T. brucei*). Induced ROS stress could then further perpetuate effects on DNA synthesis (side arrow).

DNA effects was drug specific and dependent on the specific outcomes of NTR prodrug activation, which have not been elucidated for fexinidazole.

In essence, each of the nitroaromatic drugs investigated resulted in cell cycle defects that preceded cell death by at least 24 hours. Each of these defects are different in ways that could be dissected mechanistically. Here, we focused on the loss of the S phase population during FEX treatment (Fig 2), which was shown to decrease DNA synthesis (Fig 4). It is interesting to consider what happens to these parasites if they exit S phase. Unlike mammalian cell cycle, DNA damage is not always a signal to stop cell cycle progression in trypanosomes. Parasites harboring DNA damage can have a variety of outcomes. For instance, parasites can form mass multinucleated cells with 4N or greater DNA content, which was not seen for these nitroaromatic drugs. Alternatively, DNA damage can result in asymmetric cytokinesis events that leave a large $G_2$ population of 2N cells and an increasing anucleated cell population. FEX treatment at 70 µM increases the $G_2$ population and anucleated cell population, which may suggest asymmetric cell division (Fig 2C). In addition, immunofluorescence microscopy showed that FEX treated cells have a marked increase in 2N2K ($G_2$) type cells (Fig 3), which when visualized appear to have nuclei of aberrant size and shape (S4 Fig). Collectively, these data may suggest that as fexinidazole increases DNA damage (Fig 6) and decreases DNA synthesis (Fig 4) the resulting parasites have damaged nuclei that either present as a $G_2$ stall or result in asymmetric division and an anucleated daughter. Additional experiments will be required to investigate these outcomes in more detail.

The collection of fexinidazole induced cytotoxic outcomes reported here raise the question - what comes first inhibition of DNA synthesis, DNA damage formation, or ROS formation? The induction of ROS stress, previously implicated for nitro-drugs, can result in specific molecular sources of DNA damage, including guanine oxidation resulting in 8-oxoG formation [37]. In the American trypanosome, *T. cruzi*, 8-oxoG is a common DNA lesion arising from oxidative stress and these parasites harbor a specific DNA glycosylase that has been shown to remove 8-oxoG from DNA [38,39]. A homolog of the 8-oxoG glycosylase encoding gene (*Tb927.4.2480*) is also present in the *T. brucei* genome but the pathway has not been elucidated in African trypanosomes. Thus, it is possible that fexinidazole activates ROS thereby inducing DNA damage that may be the source of DNA synthesis inhibition (Fig 8 Model A).

One limitation of the studies herein was the fact that we were only able to detect ROS stress after 24 hours using H2DCFDA (Fig 5). In contrast, fexinidazole induced DNA damage formation (Fig 6) was significantly increased after 3 hours and DNA synthesis was significantly reduced after 6 hours of treatment (Fig 4). While it is intriguing to consider the possibility that FEX induced DNA damage is the precipitating event (Fig 8 Model B), further research using more sensitive assays and synchronized cell populations are required to test this model. The data presented here show that ROS, DNA damage formation, and DNA synthesis inhibition are the outcomes of FEX treatment of *T. brucei* parasites, however elucidating the order of events will require more nuanced investigations. Notably, in other systems, not only can ROS result in DNA damage, conversely DNA damage can also result in ROS formation [40]. In mammals, DNA damage induced phosphorylation of H2AX (the γH2A-Phos homolog) results in the activation of Rac1/Nox1 (a mitochondrial NADPH oxidase) to cause persistent formation of ROS stress, which can further induce DNA damage and result in cell apoptosis [41]. To date no such pathway has been elucidated in *T. brucei* and determining if DNA damage can induce ROS stress will be the subject of future research.

Approval of the clinical use of fexinidazole was, in part, initiated to replace the use of melarsoprol in the treatment of second stage (CNS) African trypanosome infections [1]. Melarsoprol is an arsenical compound that was the sole drug available to treat second stage infections with both *T. b. gambiense* and *T.b. rhodesiense* for many decades, yet it was burdened with high levels of host toxicity [42]. The primary intracellular target of bioactivated melarsoprol (melarsen oxide) is trypanothione, a trypanosomatid specific form of glutathione [42]. In a previous publication, we showed that the primary effect of melarsoprol treatment on *T. brucei* in vitro was the complete inhibition of DNA synthesis [22]. Because melarsoprol inhibits thiol metabolism, we proposed that parasites were no longer able to provide the necessary reducing equivalents to maintain the dNTP pool (the rate limiting molecules of DNA synthesis) resulting in a complete inhibition of DNA synthesis within 24 hours [22]. It is notable that impact on cell cycle progression was different in melarsoprol than those observed for fexinidazole here, namely that melarsoprol treatment resulted in a complete loss of S phase and $G_2$ population within 24 hours [22]. Thus, we propose that while the well-established drug melarsoprol and the newly released drug fexinidazole both target DNA synthesis to effectively kill *T. brucei* parasites, this outcome likely arises through profoundly different mechanisms. Specifically, research suggest that melarsoprol blocks DNA synthesis via trypanothione inhibition (likely resulting from a depleted dNTP pool) [22]. In contrast, the findings herein suggest that DNA synthesis inhibition by fexinidazole may be precipitated by DNA damage formation. Alternatively, fexinidazole treatment may cause bulky additions to nucleic acids that both inhibit DNA synthesis and cause damage DNA resulting in some interplay between these two outcomes. Collectively our findings suggest that DNA synthesis inhibition is a powerful target of anti-trypansomatid drug therapies that could be the focus of future anti-parasitic drug development.

## Supporting information

**S1 Table. Comparison of drug concentrations table.**
(TIFF)

**S1 Data. Raw hemocytometry counts from all experiments.**
(XLSX)

**S2 Data. All flow cytometry data from cell cycle analyses.**
(XLSX)

**S3 Data. All flow cytometry data from BrdU incorporation.**
(XLSX)

**S4 Data. All flow cytometry data from H2DCFDA analysis of ROS.**
(XLSX)

**S5 Data. All flow cytometry data from γH2A-Phosphorylation analysis.**
(XLSX)

**S6 Data. All flow cytometry data from γH2A-Phosphorylation analysis.**
(XLSX)

**S1 Fig. Measurement of nitroaromatic drug EC$_{50}$.** Cell viability assays were conducted using AlamarBlue alongside Puromycin control for 100% cell death to measure the EC$_{50}$ of Benznidazole (A), Nifurtimox (B), and Fexinidazole (C). Values show drug concentrations (μM) and standard deviations resulting parasite death.
(TIFF)

**S2 Fig. Flow cytometry gating strategy for cell cycle analysis.** Representative data is shown for cell cycle gating for untreated and HIGH drug concentrations after 48 hours for a single replicate (All replicate data can be found in S2 Data. First Row displays the "Cell Gate", isolation of the cell associated events. Second Row displays "Doublet Discrimination", removal of doublet cells from data analysis such that only singlets are evaluated for cell cycle events. Third Row shows cell cycle gates for each population analyzed: Anucleated, G1, S phase, G2, and Multinucleated cell populations were quantified based on these gates. The fourth and final row shows alternative overlay visualizations of the resulting gated cell cycle data.
(TIFF)

**S3 Fig. Statistical significance of drug induced cell cycle changes.** Drug effects on G$_1$ (top), S phase (middle), and G$_2$ are shown for nifurtimox (light blue and dark blue), benznidazole (light green and dark green), and fexinidazole (yellow and orange) over 24 hours (Left) or 48 hours (Right) of drug treatment in comparison with untreated (UT) control. Cell cycle population changes of statistical significance are shown on the table and marked with asterisks according to their magnitude of significance, P values *<0.05, **<0.005, ***<0.0005.
(TIFF)

**S4 Fig. Untreated and Fexinidazole Treated Parasites.** The distribution of kinetoplasts and nuclei per parasite are shown by immunofluorescence microscopy using DAPI (DNA content staining) and anti-VSG-2-Alexa488 conjugated antibody for the parasite cell surface. Untreated presents a sample of the normal distribution of 1K1N (G$_1$) cells and 2K1N (G$_2$) cells in normal parasite populations. Fexinidazole treated (20 μM at 48 hours) parasites illustrated "other" cells types as nuclear anomalies arising from fexinidazole treatment. Bottom – Table of percent populations arising from all conditions, treated and untreated.
(TIFF)

**S5 Fig. Flow cytometry gating strategy for BrdU Incorporation Assays.** Representative data is shown for BrdU incorporation assay gating for untreated and HIGH drug concentrations after 6 hours for a single replicate (All replicate data can be found in S3 Data. First Row displays the "Cell Gate", isolation of the cell associated events. Second Row displays "Doublet Discrimination", removal of doublet cells from data analysis such that only singlets are evaluated for cell cycle events. Third Row shows quantification of the DNA synthesizing population based on staining with the anti-BrdU antibody

conjugated with Alexa-647 (y-axis) vs. DAPI DNA content staining (x-axis). The +BrdU population gate shown is the source of percent DNA synthesis used in the analyses of Fig 4. On the right a representative overlay of the data is shown.
(TIFF)

**S6 Fig. Flow cytometry gating strategy for H2DCFDA analysis of ROS.** Representative data is shown analysis of ROS by H2DCFDA staining showing a single replicate of Untreated, H2O2 positive control treatment, FEX µ20, and FEX 50 µM. First Row displays the "Cell Gate", isolation of the cell associated events. Second Row displays live vs. dead discrimination using SYTOX RED fluorescence on the y-axis. Live cells are gated (with dead cells removed) and then used to evaluate H2DCFDA under shown conditions. The "ROS+" percentage was gated based on H2DCFDA stained but Untreated cells (that is not ROS stressed). This gate was then applied to all treated samples. Overlays are shown of H2DCFDA histograms for untreated and treated conditions.
(TIFF)

**S7 Fig. Flow cytometry gating strategy for γH2A-Phosphorylation analysis of DNA Damage.** Representative data is shown analysis of DNA damage by γH2A-Phosphorylation using a *T. brucei* specific anti-γH2A-Phos antibody fused to Alexa-488. Gating shows a single replicate of Untreated and each drug at their HIGH concentrations after 3 hours. First Row displays the "Cell Gate", isolation of the cell associated events. Second Row displays Doublet discrimination based on DAPI staining for the identification and subsequent analysis of the Singlet population. In the third row singlets are plotted against anti-γH2A-Phos-A488 (y-axis) for DNA damage vs. DAPI content stating (x-axis) for evaluation of cell cycle stge. Overlays are shown for each treatment compared to untreated by countour plot as well as all drug treatments overlayed.
(TIFF)

**S8 Fig. Quantitative western blot analysis of anti-γH2A-Phos reporting of drug induced DNA damage.** Quantitation of anti-γH2A-Phos western blots normalized to anti-Tubulin control (bottom lane) for benznidazole (180 µM), fexinidazole (50 µM), and nifurtimox (15 µM) over 3, 6, 12, and 24 hours in comparison with untreated (shown as Phleo-) and phleomycin treated positive control (Phleo+).
(TIFF)

**S9 Fig. Flow cytometry gating of percent DNA damage per cell cycle stage.** Representative data from UT, BLE treated, and two replicates of each NIF 15, BENZ 180, and FEX 50 after 12 hours of treatment are shown. Data shown on the flow cytometry plot include – percent total gH2A+ (DNA damaged cells) shown above the plot, percent of events gated in G1, S, and G2. Tables below each plot show the number of total events and DNA damage specific events arising in each stage of cell cycle analyzed. The percent of DNA damage from each stage of cell cycle is derived as the number of gH2A+ events divided by the total number of events in that cell cycle stage. Full data for all conditions and replicates can be found in S6 Data.
(TIFF)

## Acknowledgments

The members of the Hovel-Miner lab would like to acknowledge the excellent training and resources provided by the GWU SMHS Flow Cytometry Core, namely Dr. Gregory Cresswell, and the training resources and equipment provided by the GW Nanofabrication & Imaging Center, specifically Dr. Anastas Popratiloff whose training was invaluable. The lab wants to extend a special thank you to Paprika Berry who was an excellent lab member during the completion of these studies.

## Author contributions

**Conceptualization:** Kenna E. Berg, Indea Rogers, Hayley M. Ramirez, Galadriel Hovel-Miner.

**Data curation:** Kenna E. Berg, Hayley M. Ramirez, Galadriel Hovel-Miner.

**Formal analysis:** Kenna E. Berg, Indea Rogers, Hayley M. Ramirez, Julian Cornejo, Galadriel Hovel-Miner.

**Funding acquisition:** Galadriel Hovel-Miner.

**Investigation:** Kenna E. Berg, Indea Rogers, Hayley M. Ramirez, Julian Cornejo, Ignacio M. Durante.

**Methodology:** Kenna E. Berg, Galadriel Hovel-Miner.

**Project administration:** Kenna E. Berg, Indea Rogers, Hayley M. Ramirez, Galadriel Hovel-Miner.

**Resources:** Galadriel Hovel-Miner.

**Supervision:** Galadriel Hovel-Miner.

**Validation:** Kenna E. Berg, Indea Rogers, Hayley M. Ramirez, Julian Cornejo, Galadriel Hovel-Miner.

**Visualization:** Kenna E. Berg, Indea Rogers, Hayley M. Ramirez, Julian Cornejo, Galadriel Hovel-Miner.

**Writing – original draft:** Galadriel Hovel-Miner.

**Writing – review & editing:** Kenna E. Berg, Indea Rogers, Hayley M. Ramirez, Julian Cornejo, Galadriel Hovel-Miner.

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
