## [Decision Letter · Decision Letter 0]

25 Jul 2025

Fexinidazole results in specific effects on DNA synthesis and DNA damage in the African trypanosome

Dear Dr. Hovel-Miner,

Thank you for submitting your manuscript to PLOS Neglected Tropical Diseases. After careful consideration, we feel that it has merit but does not fully meet PLOS Neglected Tropical Diseases's publication criteria as it currently stands. Therefore, we invite you to submit a revised version of the manuscript that addresses the points raised during the review process.

Please submit your revised manuscript within 60 days Sep 23 2025 11:59PM. If you will need more time than this to complete your revisions, please reply to this message or contact the journal office at plosntds@plos.org. Please include the following items when submitting your revised manuscript:

We look forward to receiving your revised manuscript.

Kind regards,

Brice Rotureau, PhD

Academic Editor

Susan Madison-Antenucci

Section Editor

Shaden Kamhawi

co-Editor-in-Chief

Paul Brindley

co-Editor-in-Chief

**Additional Editor Comments (if provided):**

**Journal Requirements:**

**Reviewers' Comments:**

Reviewer's Responses to Questions

**Summary and General Comments**

Reviewer #1: Detailed analyses on modes of action of fexinidazole against African trypanosomes are beneficial as the drug is becoming an option for treatment of sleeping sickness. The authors focused on the drug’s effects on cell cycle, showing some differences between fexinidazole and the other nitroaromatic drugs including nifurtimox and benznidazole. Unfortunately, their conclusion in such the difference largely depends on impressions but not statistical analyses, and the authors need to show outcomes of statistical analyses not only in comparison of a treatment group with the control group but also among the treatment groups.

What is a doubling time for this parasite? According to the growth speed and as the authors state that they used BSF, the doubling time should be short, around 6 hours. This means that even during 24 hours of cultivation, the parasites undergo multiple cell cycles. A simple question is whether it is possible to have S-phase affected while still having G2-phase not much affected. Figure 3 says that FEX decreased 2K1N but increased 2K2N; where these increased 2K2N from?

Also, the reviewer experienced difficulty in interpreting data discrepancy between Figures 2 and 3. Figure 2 says that FEX treatment resulted in the decrease in G2 phase as the other drugs did. On the other hand, Figure 3 says FEX treatment increased 2K2N. It would be nice if the authors can address this point as well.

L265: ‘Treatment with either NIF or BENZ resulted in a progressive decrease in the 2K2N population over time and concentration, which corresponds with the loss of G2 cells observed by flow cytometry (FIG. 3B).’ Not sure if this interpretation is correct. For example, NIF 48 h has 14% 2K2N (G2), which is higher than that of UT.

Nonetheless, this study only did analyses on DNA incorporation and cell cycle analyses (and some on ROS). To increase the significance of the study, it may be critical to do more comprehensive analyses of the drug effect on cell cycle using synchronized parasites. There are a couple of methods proposed for cell cycle synchronization in Trypanosoma brucei. Is there any reason not introducing the methodology into this study?

L338: Why is phleomycin abbreviated as BLE?

Reviewer #2: This manuscript investigates the cytotoxic effects of fexinidazole, a clinically approved oral nitroaromatic drug, in Trypanosoma brucei, and compares its activity with that of nifurtimox and benznidazole. The authors evaluate the effects of these compounds on cell cycle progression, DNA synthesis, DNA damage (via γH2A phosphorylation), and ROS production. This work aims to dissect whether fexinidazole exerts distinct cellular effects that may inform its mechanism of action and distinguish it from related nitroaromatic agents.

Strengths of the Study:

Timely and relevant topic: Understanding the mode of action of fexinidazole is of considerable importance, given its recent approval as an oral treatment for HAT and its potential application against other kinetoplastid infections.

Comparative design: The inclusion of two other clinically used nitroaromatic drugs (nifurtimox and benznidazole) provides a useful comparative framework to evaluate the specificity of fexinidazole's effects.

Use of cell biology tools: The application of flow cytometry for cell cycle and γH2A-phosphorylation, BrdU incorporation, and ROS detection provides a multi-angle view of cytotoxic phenotypes.

Potential for broader impact: The findings, if mechanistically clarified, could influence drug development strategies for kinetoplastid diseases and contribute to our understanding of drug-induced DNA damage pathways in protozoan parasites.

Weaknesses and Major Limitations:

Despite its strengths, the manuscript falls short of providing definitive mechanistic insight and currently overinterprets several findings:

Inferred causality is not experimentally demonstrated.

The authors propose that fexinidazole-induced DNA damage precedes and drives DNA synthesis inhibition and ROS formation. However, this sequence is inferred based on asynchronous measurements across different timepoints and lacks direct experimental validation. Without synchronized assays or causal perturbations, this interpretation remains speculative.

Nature of DNA damage is not characterized.

The exclusive reliance on γH2A phosphorylation does not distinguish between types of DNA lesions (e.g., strand breaks vs. oxidative adducts). This limits the conclusions that can be drawn about the underlying mechanism.

Drug dosing strategy lacks standardization.

The use of differing multiples of EC50 across drugs (e.g., 6×EC50 for fexinidazole vs. 2×EC50 for benznidazole) compromises the comparability of phenotypic effects and may exaggerate fexinidazole’s distinctiveness.

No investigation of DNA repair or downstream pathways.

The authors mention the potential involvement of oxidative DNA repair mechanisms (e.g., 8-oxoG glycosylase) but do not experimentally assess whether these are activated or impaired during fexinidazole exposure.

Novelty and Significance:

The manuscript provides new comparative phenotypic data on three nitroaromatic drugs in T. brucei and identifies a potentially distinctive inhibitory effect of fexinidazole on DNA synthesis. The implementation of flow cytometry to measure DNA damage across cell cycle stages is technically valuable and potentially broadly applicable.

However, to achieve the level of mechanistic novelty necessary for publication in PLOS Neglected Tropical Diseases, the manuscript must go beyond observational data and test the proposed relationships between phenotypes more rigorously.

New Experiments Required for Acceptance (Major Revision):

To support the authors’ central hypothesis and improve mechanistic depth, the following experiments or analyses are strongly recommended:

Clarify the temporal relationship between DNA damage, ROS formation, and DNA synthesis inhibition:

Improve ROS detection sensitivity at earlier timepoints (e.g., 3–6 h) using more sensitive probes or alternative assays.

Optionally, use synchronized cell populations to track phenotypes more precisely through the cell cycle.

Characterize the nature of DNA damage:

Use comet assay, γH2A foci visualization, or biochemical approaches (e.g., LC-MS for DNA adducts) to determine whether damage is oxidative, alkylative, or strand break-associated.

Standardize drug dosing:

Compare drugs at equitoxic doses (e.g., IC90 or AUC-equivalent) or provide dose-response curves that justify the specific concentrations used for each drug.

Investigate DNA repair pathways:

Analyze expression or activity of candidate DNA repair genes (e.g., glycosylases, polymerases) in fexinidazole-treated vs. untreated cells, or assess parasite viability in DNA repair-deficient strains.

If these experiments cannot be completed within the revision timeline, the authors must significantly temper their conclusions and explicitly frame the mechanistic model as speculative.

Scholarship and Ethical Standards:

The manuscript appears to meet scientific and ethical standards. No concerns were identified regarding research misconduct, dual publication, or data transparency. The data availability statement should be updated with links to raw data (e.g., flow cytometry files) upon revision to support reproducibility.

Final Summary:

This study addresses an important clinical agent and provides promising comparative data. However, in its current form, the manuscript falls short of its mechanistic ambition and would benefit from deeper experimental support and more cautious interpretation. I recommend Major Revision, with the expectation that the authors either (1) perform key mechanistic experiments or (2) revise their claims to align with the descriptive nature of the data.

Reviewer #3: The authors address a critical and recent topic regarding HAT treatment, demonstrating in vitro the possible mechanisms of action of three drugs, focusing on fexinizadole, a drug approved by DNDi for the oral treatment of African trypanosomiasis. The text and results are clear, corroborating most of the proposed findings. However, there are some improvements that could adapt the study to the journal format and facilitate the understanding of the manuscript.

Suggestions:

1) Manuscript organization: I suggest that authors review the “manuscript organization” section as authors of research articles are asked to add an Author summary after the Abstract.

- I suggest that authors review the format requested by the journal for figure labels and in-text citations throughout the text and in the figure captions.

2) Introduction:

The introduction is clear and objective, but I suggest the authors discuss African trypanosomiasis in more detail, providing recent data on its epidemiology, number of cases, and endemic regions, since the article's focus seems to be more related to HAT than to other trypanosomiasis.

Line 113: Add a reference to the text about HAT.

3) Results:

Lines 193-202: In the excerpt, the authors use the terms "low" and "high" to discuss the concentrations of nitroaromatic drugs. However, it's unclear whether when they refer to all drug concentrations (low and high), they are referring to Figure 1B or 1C (or both). It's in Figure C that the authors show the graphs of low nitro compared and high nitro compared (and cite in the figure caption as "more detailed visualization of data show in panel (B)"). I suggest the authors review this information and, if they find it pertinent, add to the excerpt that the results presented are in Figures 1B-C.

Lines 246-251: I suggest the authors add a reference at the end of the paragraph.

Fig 6: The authors show the overlay of the different drugs in the last graphs of Figure 6A, but I believe an overlay using a contour plot would be more visual and highlight the results.

Lines 366-367:

- I suggest the authors add to the legend of Fig. 6 the statistical analysis used in graph 6C (t-test, one-way or two-way ANOVA, or others).

Fig 6D: Is there any reason the authors used flow cytometry at 3, 6, and 12 hours and Western blot at 3, 6, 12, and 24 hours? Do the authors expect that the DNA damage induced by the drugs, especially fexinidazole, would be greater at 24 hours?

Line 375: "... (FIG. 7B, note the maximum scale of graph G1)." I suggest the authors review whether the journal allows this type of configuration for in-text citations.

Fig 7: I suggest the authors add in the legend the statistical analysis used in graphs A and C (t-test, one-way or two-way anova, or others).

- The authors, based on yH2A-Phos and DAPI staining, quantified the proportion of DNA damage in each phase of the cell cycle, adding multinucleated cells to the last quadrant. Why evaluate these cells? Would this evaluation indicate that the drug was interfering with cell cycle progression? Is there a statistical difference from this data in Figure 7C (Multinuclear damage)?

- I suggest the authors add the yH2A-Phos and DAPI subgates with the NIF, BENZ, and FEX treatments to complement the data shown in Fig 7C (Multinuclear DNA damage).

4) Methods:

Lines 495-528: I suggest authors review this Methods section, as despite containing all the relevant information, the text is lengthy and somewhat confusing. I suggest dividing it into more sections could improve reader understanding.

Line 498: The authors cite the drug eflornithine, but it is not mentioned throughout the paper, only in the discussion because it is a combination therapy with nifurtimox in humans. Was this drug used? If not, I suggest the authors remove it from the Methods.

Lines 530-538: I suggest that the authors remove the statistical information from this topic and add a new topic “Statistical analysis” so that they can add information on which statistical analyses were used, which tool (e.g. Graph Pad Prism) and which p value was considered statistically significant.

Line 535: I suggest that the authors add which version of FlowJo was used.

Line 542: Since this is the first time the authors have cited "PBS," I suggest writing out the solution name.

Lines 555-556: I suggest the authors add the centrifugation speed (rpm).

5) References:

Line 845: I suggest that the authors review the reference, as it was repeated in (1) (line 827).

PLOS authors have the option to publish the peer review history of their article (what does this mean? ). If published, this will include your full peer review and any attached files.

**Do you want your identity to be public for this peer review?** For information about this choice, including consent withdrawal, please see our Privacy Policy .

Reviewer #1: No

Reviewer #2: No

Reviewer #3: No

**Key Review Criteria Required for Acceptance?**

**Methods**

-Are the objectives of the study clearly articulated with a clear testable hypothesis stated?

-Is the study design appropriate to address the stated objectives?

-Is the population clearly described and appropriate for the hypothesis being tested?

-Is the sample size sufficient to ensure adequate power to address the hypothesis being tested?

-Were correct statistical analysis used to support conclusions?

-Are there concerns about ethical or regulatory requirements being met?

Reviewer #2: 1. Are the objectives of the study clearly articulated with a clear testable hypothesis stated?

The overall objective of the manuscript is relevant and timely—namely, to delineate the specific cytotoxic effects of fexinidazole in T. brucei and to distinguish these from other nitroaromatic drugs. However, the hypothesis that "fexinidazole-induced DNA damage precedes and drives ROS formation and DNA synthesis inhibition" is only partially supported by the current data and not fully testable in its present form.

Required Revision:

The authors should clearly articulate their hypothesis in a testable format and either:

(a) refine it to reflect the current scope of their data (i.e., correlation rather than causation), or

(b) include new data (e.g., earlier ROS detection, kinetic comparison) that would allow a more direct test of the proposed sequence of events.

2. Is the study design appropriate to address the stated objectives?

The authors used relevant phenotypic assays (BrdU incorporation, γH2A phosphorylation, ROS detection) and comparative drug exposure models to examine cytotoxicity. However, there are notable gaps in the mechanistic evaluation, particularly the absence of data clarifying the nature of DNA damage and the incomplete assessment of repair responses.

Required Revision:

To address the mechanistic objective of the study, additional assays are needed to define the nature of DNA damage (e.g., comet assay, adduct detection) or repair activity. Alternatively, the authors must adjust the scope and interpretations of their claims to align with their descriptive findings.

3. Is the population clearly described and appropriate for the hypothesis being tested?

Yes. The use of Trypanosoma brucei bloodstream form (Lister427, SM line) is appropriate and well-justified for studying fexinidazole’s effects in the context of HAT.

4. Is the sample size sufficient to ensure adequate power to address the hypothesis being tested?

The study claims to use three or more biological replicates in most assays. However, statistical analysis details are limited, and it's not clear whether power calculations or tests for significance across multiple comparisons were applied correctly.

Required Revision:

Authors should provide detailed information on sample sizes for each experiment, clarify whether replicates were technical or biological, specify the statistical tests used, and apply corrections for multiple comparisons where appropriate.

5. Were correct statistical analyses used to support conclusions?

Statistical significance is reported, but the manuscript lacks sufficient detail regarding the tests used, assumptions checked (e.g., normality), and whether any multiple hypothesis correction was applied.

Required Revision:

The authors must clearly describe statistical methods in both the figure legends and the Methods section, including assumptions, test types, software used, and correction methods for multiple comparisons.

6. Are there concerns about ethical or regulatory requirements being met?

No concerns are noted. The study involves in vitro parasite culture only, with no vertebrate or human subjects. Ethical standards appear to have been met.

Summary

While the study provides valuable comparative insights into nitroaromatic drug effects in T. brucei, it falls short of providing mechanistic clarity sufficient for publication without substantial revision. Addressing the experimental and interpretive limitations detailed above is required for acceptance.

Reviewer #3: The methods section is clear and well-formatted. However, some minor changes are suggested below.

**Results**

-Does the analysis presented match the analysis plan?

-Are the results clearly and completely presented?

-Are the figures (Tables, Images) of sufficient quality for clarity?

Reviewer #2: 1. Does the analysis presented match the analysis plan?

Yes, the analyses align with the general objectives stated in the Introduction. The authors aimed to compare the effects of fexinidazole, benznidazole, and nifurtimox on T. brucei cell cycle progression, DNA synthesis, DNA damage, and ROS production. The study follows a logical flow of experiments from drug titration and viability assays to mechanistic phenotyping.

However, given the stated goal to uncover drug-specific mechanisms, the current analyses remain largely descriptive. The mechanistic implications discussed (e.g., DNA damage initiating ROS and DNA synthesis inhibition) are not directly tested or planned for in the methodology, which creates a mismatch between the depth of analysis and the interpretation.

Suggestion: The authors should either (a) temper their mechanistic claims in the discussion, or (b) provide additional experiments to directly test the sequence of events hypothesized.

2. Are the results clearly and completely presented?

The results are generally well organized and presented in a logical sequence. Each experimental section is supported by descriptive narrative and corresponding figures. However, the following points require attention:

Some key experimental parameters (e.g., exact sample sizes per experiment, gating strategy details, biological vs. technical replicates) are missing or buried in supplementary material;

Statistical results are incompletely reported—many comparisons lack clear p-values, test types, or error bar definitions;

Conclusions occasionally go beyond what the data directly show, especially in attributing causality between phenotypes.

Suggestion: The clarity and rigor of the Results section would benefit from enhanced statistical transparency and more cautious language where mechanistic inference is not directly supported.

3. Are the figures (Tables, Images) of sufficient quality for clarity?

Figures are generally of good technical quality and include appropriate biological replicates. Flow cytometry plots, histograms, and violin plots are effectively used to convey quantitative data. However:

Some flow cytometry figures (e.g., Figures 6–7) would benefit from more explicit axis labeling and gating overlays;

Figure legends should include full definitions of abbreviations, concentrations, and the number of replicates;

Figure 8 (mechanistic model) should more clearly indicate which pathways are data-driven vs. speculative.

Suggestion: Improve figure annotations and ensure all necessary explanatory details are available in the legends to make the figures fully self-contained.

Reviewer #3: The results, figures, and analyses are clear. However, minor observations and suggestions are made below.

**Conclusions**

-Are the conclusions supported by the data presented?

-Are the limitations of analysis clearly described?

-Do the authors discuss how these data can be helpful to advance our understanding of the topic under study?

-Is public health relevance addressed?

Reviewer #2: 1. Are the conclusions supported by the data presented?

The authors conclude that fexinidazole exerts a distinct trypanocidal effect through early and pronounced inhibition of DNA synthesis, along with DNA damage accumulation and ROS formation. While their experimental data clearly show differences in cell cycle profiles, DNA synthesis (via BrdU), and γH2A phosphorylation compared to other nitroaromatic drugs, the claim of a unique mechanistic sequence—where DNA damage precedes and drives the other effects—is only partially supported.

In particular, the evidence does not definitively establish causality or sequence between DNA damage, ROS, and DNA synthesis inhibition. The timepoints measured for ROS and DNA damage differ in resolution, and the mechanism of damage (e.g., oxidative vs. alkylation) is not defined. Therefore, certain conclusions are overstated relative to the data.

Recommendation: The authors should temper their mechanistic claims and avoid inferring causality without direct supporting experiments.

2. Are the limitations of analysis clearly described?

The manuscript acknowledges certain limitations, including the lack of knowledge regarding the chemical nature of fexinidazole’s active metabolites and the possibility of cross-resistance via NTR mutations. However, important analytical limitations are not fully addressed, such as:

The inability to detect early ROS events due to assay limitations;

The lack of specificity of γH2A as a DNA damage marker;

The absence of DNA repair analysis or metabolite profiling.

Recommendation: The authors should expand the discussion of limitations to provide a more balanced perspective and clarify where further work is needed.

3. Do the authors discuss how these data can be helpful to advance our understanding of the topic under study?

Yes. The authors correctly highlight the need for better understanding of nitroaromatic drug mechanisms, especially for fexinidazole, which is now a first-line oral treatment for HAT. They point out that DNA synthesis inhibition could represent a key vulnerability in T. brucei, and that differentiating the cytotoxic effects of individual drugs may inform future drug design.

This is a useful and important perspective, particularly given the limited pipeline for kinetoplastid-targeting therapeutics and growing concerns about drug resistance.

4. Is public health relevance addressed?

Yes, the clinical importance of fexinidazole is discussed in the context of replacing melarsoprol, a highly toxic arsenical drug, for second-stage HAT. The manuscript also notes the potential of fexinidazole for broader use against other kinetoplastids, including Leishmania and T. cruzi. These points are appropriately highlighted in the Introduction and Discussion.

That said, the link between the current molecular findings and broader public health implications could be made stronger. For example, how might these mechanistic insights inform drug resistance monitoring in endemic settings? Can DNA damage be used as a biomarker of treatment efficacy or toxicity?

Suggestion: Strengthen the bridge between mechanistic insights and public health applications in the concluding paragraphs.

Reviewer #3: The results supported most of the conclusions and are relevant for research. Some observations were adressed below.

**Editorial and Data Presentation Modifications?**

Reviewer #2: The manuscript is generally well written and clearly structured. However, a number of editorial and presentation-related revisions are recommended to improve clarity, consistency, and readability. These suggestions are relatively minor and do not affect the scientific content but will enhance the communication and transparency of the findings.

1. Language and Style:

Some sections of the Discussion include speculative or overly definitive statements (e.g., "DNA damage precedes and drives ROS and DNA synthesis inhibition"). These should be revised for greater caution and scientific neutrality (e.g., “may contribute to” or “is temporally associated with”).

Avoid redundancy in phrase structures (e.g., “the results show that fexinidazole treatment results in…” could be simplified).

Define abbreviations consistently upon first use (e.g., gH2A-P, BrdU, ROS) and ensure they are used uniformly throughout the text.

Minor grammatical and syntactic edits are needed in some figure legends and method descriptions.

2. Figure Improvements:

Figures 6 & 7 (Flow Cytometry):

Improve axis labeling (e.g., fluorescence units, DAPI intensity), and clearly mark gating thresholds used to define positive populations.

Include representative dot plots or histograms for key time points, ideally with overlays of untreated and positive control (phleomycin-treated) for visual comparison.

Figure 8 (Proposed Model):

Distinguish more clearly between experimentally supported pathways and hypothetical ones. For example, use dashed arrows for proposed links and solid arrows for supported mechanisms. Explicitly label speculative pathways as “putative.”

Figure Legends:

Provide full details in legends so that figures are interpretable independently of the main text. This includes specifying drug concentrations, treatment durations, number of replicates, statistical methods (e.g., “mean ± SD from n=3 biological replicates”), and significance thresholds.

3. Data Clarity and Statistics:

Specify in the text and figure legends whether error bars represent standard deviation (SD) or standard error of the mean (SEM).

Clarify the statistical tests used for each dataset, including whether multiple comparisons were adjusted and whether data distribution assumptions were tested.

Consider presenting selected summary data in tabular form in Supplementary Materials for transparency (e.g., % BrdU-positive cells at each time point for all drugs).

4. Supplemental Data:

The manuscript refers to multiple supplemental figures, but it would be helpful to include a supplemental table summarizing EC50, IC90, and final concentrations used for each drug to facilitate direct comparison.

Consider including raw flow cytometry plots and gating strategies as supplementary figures for reproducibility.

5. Data Availability and Transparency:

The data availability statement should include a DOI or repository link for raw data (e.g., flow cytometry files, western blots), especially for key experiments involving gating and quantification.

Summary:

These editorial and presentation-related suggestions are intended to improve the clarity, transparency, and visual quality of the manuscript. None of the above issues warrant rejection or invalidate the conclusions but are important for enhancing scientific communication and reproducibility.

If the authors implement these modifications alongside the more substantive revisions requested, the manuscript will be significantly improved in clarity and presentation.

**Figure resubmission:**

**Reproducibility:**



---

## [Decision Letter · Decision Letter 1]

25 Sep 2025

Fexinidazole results in specific effects on DNA synthesis and DNA damage in the African trypanosome

Dear Dr. Hovel-Miner,

Thank you for submitting your manuscript to PLOS Neglected Tropical Diseases. After careful consideration, we feel that it has merit but does not fully meet PLOS Neglected Tropical Diseases's publication criteria as it currently stands. Therefore, we invite you to submit a revised version of the manuscript that addresses the points raised during the review process.

Please submit your revised manuscript within 60 days Oct 25 2025 11:59PM. If you will need more time than this to complete your revisions, please reply to this message or contact the journal office at plosntds@plos.org. Please include the following items when submitting your revised manuscript:

We look forward to receiving your revised manuscript.

Kind regards,

Brice Rotureau, PhD

Academic Editor

Susan Madison-Antenucci

Section Editor

Shaden Kamhawi

co-Editor-in-Chief

Paul Brindley

co-Editor-in-Chief

**Additional Editor Comments :**

Please, carefully consider all comments from all reviewers and provide clear answers to their questions.

Some statistical analyses and plots should especially be improved, and the interpretations of Figure 2 and Figure 3 clarified.

**Journal Requirements:**

1) Please amend your detailed Financial Disclosure statement. This is published with the article. It must therefore be completed in full sentences and contain the exact wording you wish to be published.

1) State what role the funders took in the study. If the funders had no role in your study, please state: "The funders had no role in study design, data collection and analysis, decision to publish, or preparation of the manuscript.".

**Reviewers' Comments:**

Reviewer's Responses to Questions

**Summary and General Comments**

Reviewer #1: I believe that all the three reviewers pointed out the necessity of more mechanistic studies and unless otherwise the manuscript would not be worthy for publishing in PLOS NTDs. Now I found that the authors refused to do so.

In addition, the quality of revisions/responses to reviewers’ comments is also not very satisfactory. So I regret to say that the manuscript does not reach to a standard level for publication.

Figure 2B: It is not appropriate to compare the four groups one-by-one by picking two out of the four groups. The current form has problems by showing the identical data in separate graphs and by choosing inappropriate statistical analyses.

I asked about decreased 2K1N and increased 2K2N. Because 2K2N follows after 2K1N, if 2K1N is forced to decrease, the following 2K2N should decrease too in a simple model. Another model to explain decreased 2K1N and increased 2K2N would be an accelerated/shortened cycle of 2K1N. Nonetheless, the authors failed to comprehensively discuss their data again.

I also pointed out the discrepancy between Figure 2 and Figure 3 in the original article, as the original Figure 2 showed rather decreased G2 phase (2K2N) while Figure 3 showing increased 2K2N. Then the authors just hid the data on G2 phase in Figure 2B. But unfortunately, the flow data in Figure 2A still says the proportion of G2 is rather decreased by FEX treatment and the trend is completely opposite from one shown in Figure 3B.

I also mentioned ‘For example, NIF 48 h has 14% 2K2N (G2), which is higher than that of UT’ and a response from the authors was ‘19% on UT is greater than 14% on NIF 48h’. It is sad that they do not understand the difference between absolute numbers and percentages. The numbers in the bar graph are absolute numbers but not percentages, and the authors should compare percentages, i.e., 19/154 =12% on UT and 14/102 = 14% on NIF.

Reviewer #2: This is a well-conceived and carefully executed mechanistic study that fills a clear gap by comparing three frontline nitro drugs—nifurtimox, benznidazole, and fexinidazole—side-by-side in bloodstream Trypanosoma brucei. The work is timely and relevant to human African trypanosomiasis (HAT), and the manuscript is clearly written, logically structured, and easy to follow.

Novelty & significance: A genuine head-to-head evaluation that delineates a distinct functional signature for fexinidazole (relative to nifurtimox/benznidazole), advancing our understanding of nitro-drug action and offering hypotheses for regimen optimization or combinations in HAT.

Sound design & orthogonal readouts: A coherent pipeline (dose–response and growth/death kinetics to define sublethal windows → BrdU for DNA synthesis → γH2A for DNA damage → ROS assays → cell-cycle profiling) with appropriate positive/negative controls and live/dead discrimination, providing convergent mechanistic evidence.

Rigor & reproducibility: Methods are detailed (reagents, instruments, software), gating strategies are transparent, and replicate structure and time-course design enable consistent interpretation across assays.

Presentation quality: Figures are clear and well annotated; the narrative maintains excellent alignment between aims, methods, results, and conclusions, with a balanced discussion that situates the findings in current literature.

Public-health relevance: The conclusions explicitly connect mechanistic differences to therapeutic decision-making in HAT (e.g., informing dosing windows, potential biomarkers, and resistance monitoring).

Ethics/compliance: Entirely in vitro; no concerns regarding research or publication ethics. No evidence of dual publication.

Reviewer #3: The authors made the suggested corrections, with the exception of those listed below. I suggest reviewing these points to further enhance the quality of the publication.

Figure 6 and 7: I suggest that the authors add which statistical test was used (t-test/anova) in the figure legend.

References: I suggest that authors review the list of references, as ref Deeks ED. Fexinidazole: First Global Approval. Drugs 2019, 79:215-20, remains duplicated on lines 848 and 876.

PLOS authors have the option to publish the peer review history of their article (what does this mean? ). If published, this will include your full peer review and any attached files.

**Do you want your identity to be public for this peer review?** For information about this choice, including consent withdrawal, please see our Privacy Policy .

Reviewer #1: No

Reviewer #2: No

Reviewer #3: No

**Key Review Criteria Required for Acceptance?**

**Methods**

-Are the objectives of the study clearly articulated with a clear testable hypothesis stated?

-Is the study design appropriate to address the stated objectives?

-Is the population clearly described and appropriate for the hypothesis being tested?

-Is the sample size sufficient to ensure adequate power to address the hypothesis being tested?

-Were correct statistical analysis used to support conclusions?

-Are there concerns about ethical or regulatory requirements being met?

Reviewer #2: The study is methodologically sound and addresses an important mechanistic question in trypanocidal pharmacology. With the required statistical refinements, explicit hypothesis statement, clearer dose/time selection rationale, and minimal reporting upgrades for cytometry and EC50 modeling, the manuscript will meet the journal’s criteria for methodological rigor.I recommend minor revision.

State a clear, testable hypothesis and link it to pre-specified endpoints.

Strengthen statistics and reporting.

Replace multiple t-tests with one/two-way ANOVA or a suitable mixed-effects model plus multiplicity correction (Holm–Šidák/Tukey/FDR); report exact P, effect sizes, and 95% CIs in legends (consider AUC for time-courses).

Clarify dose/time selection and cytometry/EC50 details.

**Results**

-Does the analysis presented match the analysis plan?

-Are the results clearly and completely presented?

-Are the figures (Tables, Images) of sufficient quality for clarity?

Reviewer #2: Please export plots and flow histograms as vector graphics (PDF/SVG/EPS) and provide micrographs at their native resolution in TIFF/PNG (≥300 dpi at final print size; ≥600 dpi for line art). Avoid upscaling raster images in Photoshop or similar software, as it cannot recover detail and may introduce artifacts.

**Conclusions**

-Are the conclusions supported by the data presented?

-Are the limitations of analysis clearly described?

-Do the authors discuss how these data can be helpful to advance our understanding of the topic under study?

-Is public health relevance addressed?

Reviewer #2: End the Conclusions with 2–3 sentences on how these results refine understanding of nitro drug action and how that could inform HAT control (e.g., regimen optimization, rational combinations, resistance monitoring/biomarker development). Make the public-health angle explicit (impact on treatment durability and programmatic decision-making).

**Figure resubmission:**
---

## [Editor Report · Decision Letter 2]

15 Oct 2025

Dear Dr Hovel-Miner,

We are pleased to inform you that your manuscript 'Fexinidazole results in specific effects on DNA synthesis and DNA damage in the African trypanosome' has been provisionally accepted for publication in PLOS Neglected Tropical Diseases.

Best regards,

Brice Rotureau, PhD

Academic Editor

Susan Madison-Antenucci

Section Editor

Shaden Kamhawi

co-Editor-in-Chief

Paul Brindley

co-Editor-in-Chief

---

## [Editor Report · Acceptance letter]

Dear Dr Hovel-Miner,

We are delighted to inform you that your manuscript, "Fexinidazole results in specific effects on DNA synthesis and DNA damage in the African trypanosome," has been formally accepted for publication in PLOS Neglected Tropical Diseases.

Best regards,

Shaden Kamhawi

co-Editor-in-Chief

Paul Brindley

co-Editor-in-Chief
